# Quantum spin liquid signatures in monolayer 1T-NbSe$_2$

Quanzhen Zhang[1,7], Wen-Yu He [2,7], Yu Zhang [1,3,7] ✉, Yaoyao Chen[1,7], Liangguang Jia[1], Yanhui Hou[1], Hongyan Ji[1], Huixia Yang[1], Teng Zhang [1], Liwei Liu[1], Hong-Jun Gao [4], Thomas A. Jung [5] & Yeliang Wang [1,6] ✉

Quantum spin liquids (QSLs) are in a quantum disordered state that is highly entangled and has fractional excitations. As a highly sought-after state of matter, QSLs were predicted to host spinon excitations and to arise in frustrated spin systems with large quantum fluctuations. Here we report on the experimental observation and theoretical modeling of QSL signatures in monolayer 1T-NbSe$_2$, which is a newly emerging two-dimensional material that exhibits both charge-density-wave (CDW) and correlated insulating behaviors. By using scanning tunneling microscopy and spectroscopy (STM/STS), we confirm the presence of spin fluctuations in monolayer 1T-NbSe$_2$ by observing the Kondo resonance as monolayer 1T-NbSe$_2$ interacts with metallic monolayer 1H-NbSe$_2$. Subsequent STM/STS imaging of monolayer 1T-NbSe$_2$ at the Hubbard band energy further reveals a long-wavelength charge modulation, in agreement with the spinon modulation expected for QSLs. By depositing manganese-phthalocyanine (MnPc) molecules with spin $S = 3/2$ onto monolayer 1T-NbSe$_2$, new STS resonance peaks emerge at the Hubbard band edges of monolayer 1T-NbSe$_2$. This observation is consistent with the spinon Kondo effect induced by a $S = 3/2$ magnetic impurity embedded in a QSL. Taken together, these experimental observations indicate that monolayer 1T-NbSe$_2$ is a new promising QSL material.

Quantum spin liquids (QSLs) are in a novel quantum state with a variety of unusual properties that have been long sought in condensed matter physics. In a QSL material, spins of electrons are highly entangled and exhibit no magnetic order even down to the zero-temperature limit[1–4]. Due to the intrinsic quantum fluctuations, QSLs support fractionalized elementary excitations and contain emergent gauge fields[2–7]. A correlated insulator with a triangular lattice has been predicted able to host a QSL ground state under certain conditions[6–11]. For the material candidate 1T-TaS$_2$ that was firstly predicted to exhibit QSL signatures[8–11], however, the true ground state remains unclear[12–18], leaving the exact nature of QSL states in triangular lattice elusive. The search for QSL signatures in new triangular systems and the unveiling of their cooperative properties are, therefore, of great significance.

Very recently, monolayer 1T-TaSe$_2$, one of the isostructural compounds with 1T-TaS$_2$, was found to be a correlated insulator and exhibits the signatures of the U(1) QSL with spinon Fermi surface (SFS)[19,20]. Almost simultaneously, another isostructural compound, monolayer 1T-NbSe$_2$, was verified to be a newly emerging two-dimensional (2D)

[1]School of Integrated Circuits and Electronics, MIIT Key Laboratory for Low-Dimensional Quantum Structure and Devices, Beijing Institute of Technology, Beijing 100081, China. [2]School of Physical Science and Technology, ShanghaiTech University, Shanghai 201210, China. [3]Advanced Research Institute of Multidisciplinary Sciences, Beijing Institute of Technology, Beijing 100081, China. [4]Institute of Physics, Chinese Academy of Sciences, Beijing 100190, China. [5]Laboratory for X-ray Nanoscience and Technologies, Paul Scherrer Institut (PSI), 5232 Villigen, Switzerland. [6]Yangtze Delta Region Academy, Beijing Institute of Technology, Jiaxing, Zhejiang 314000, China. [7]These authors contributed equally: Quanzhen Zhang, Wen-Yu He, Yu Zhang, Yaoyao Chen. ✉e-mail: yzhang@bit.edu.cn; yeliang.wang@bit.edu.cn

correlated insulator[21-26]. Upon cooling, monolayer 1T-NbSe$_2$ undergoes a commensurate charge-density-wave (CDW) phase transition induced by a combination of electron-electron and electron-phonon interactions, resulting in a well-ordered triangular $\sqrt{13} \times \sqrt{13}$ CDW lattice with Star-of-David (SOD) motifs[22]. For monolayer 1T-NbSe$_2$ in a commensurate CDW state, there is a half-filled flat band dominated by the $d_{z^2}$-orbitals of the central Nb atom in each SOD motif. The Coulomb repulsion at the central Nb sites further splits the flat band into an upper and a lower Hubbard bands (UHB and LHB), yielding a correlated insulating state in this system[21-26]. Interestingly, monolayer 1T-NbSe$_2$ assumes a similar crystalline and electronic structure to the QSL candidates 1T-TaS$_2$ and 1T-TaSe$_2$. Therefore, it becomes important to investigate the nature of the correlated insulating state and possible QSL in triangular lattice of monolayer 1T-NbSe$_2$.

In this work, we report conclusive evidence for QSL behavior in monolayer 1T-NbSe$_2$ via scanning tunneling microscopy/spectroscopy (STM/STS) measurements and theoretical modeling. We investigate the modified density of states (DOS) in single 1T-NbSe$_2$ layers on electronically passive substrate in presence and absence of a metallic spacer-monolayer of 1H-NbSe$_2$. In presence of the metal layer, we observe periodic patterns of Kondo resonance peaks, implying that each SOD motif of monolayer 1T-NbSe$_2$ hosts one localized spin. Absence of the metal layer, STS images of the local electronic DOS at the Hubbard band energies of monolayer 1T-NbSe$_2$ reveal a long-wavelength modulation, which evidences a standing wave induced by the fractionalized quasiparticles inside the correlated insulating gap[9,19]. By depositing a magnetic molecule with $S = 3/2$ spin center onto monolayer 1T-NbSe$_2$, new resonance peaks emerge at the Hubbard band edges of monolayer 1T-NbSe$_2$, which agrees well with the spinon Kondo effect induced by a magnetic impurity deposited onto a gapless U(1) QSL[27,28]. Thereby our series of experiments demonstrate that monolayer 1T-NbSe$_2$ is a promising U(1) QSL material candidate with spinon Fermi surface (SFS).

## Results

### Quantum spin fluctuations in monolayer 1T-NbSe$_2$

We produce high-quality NbSe$_2$ films on bilayer graphene (BLG)/SiC(0001) substrates via molecular beam epitaxy (MBE) (see "Methods")[25,29]. The regions of monolayer 1T-NbSe$_2$, monolayer 1H-NbSe$_2$, and their layering on the substrate can be clearly distinguished in our STM study, as demonstrated in Fig. 1 and Supplementary Fig. 1. 1T-NbSe$_2$ can be recognized by the ordered $\sqrt{13} \times \sqrt{13}$ CDW pattern of SOD motifs at low temperatures[22-26]. Within each SOD motif, 12 surrounding Nb atoms contract towards one central Nb atom in the middle layer, accompanied by six top-layer Se atoms and six bottom-layer Se atoms. From the atomic-resolution STM image shown in the inset of Fig. 1b and Supplementary Fig. 2, we can find out that each SOD motif exhibits as a triangle, because the six top-layer Se atoms have more contribution to the STM images. 1H-NbSe$_2$, in contrast, exhibits a $3 \times 3$ CDW lattice aligned with the atomic lattice (Supplementary Fig. 3)[30].

Figure 1c shows representative STS spectra of monolayer 1T-NbSe$_2$, recorded on and off the SOD motifs, as labeled by the top and hollow sites of the CDW pattern. The DOS peaks at the energies of about −0.3 eV, 0.2 eV, and 0.6 eV are mainly attributed to the LHB hybridizing with the valence band (VB), UHB$_1$, and UHB$_2$, respectively. The splitting of the UHB into UHB$_1$ and UHB$_2$ is attributed to the spatially varying Coulomb repulsion and the reduced screening in 2D systems, which is consistent with monolayer 1T-TaSe$_2$ that was previously demonstrated[19,20]. In addition, the UHB$_2$ is highly hybridized with the UHB$_1$ because of the composite nature of the UHB (contributed by 13 Nb atoms in each SOD). It's worth noting that such a correlated insulating behavior is absent for bilayer 1T-NbSe$_2$ (actually bilayer 1T-NbSe$_2$ exhibits a metallic state, see Supplementary Fig. 4), owing to the existence of interlayer coupling.

Moreover, the intensities of these DOS peaks exhibit remarkably spatial dependency. As we can see from the spatially resolved STS spectra in Fig. 1d, the LHB is predominantly localized at the top sites of SOD motifs, while the UHB$_1$ and UHB$_2$ are at the hollow sites, in accordance with previously reported[23-26,29]. Such a result can also be intuitively presented from the orbital textures. Here we show the measured STS maps at the Hubbard band energies of monolayer 1T-NbSe$_2$ in Supplementary Fig. 5, and compare the orbital textures of monolayer 1T-NbSe$_2$ to those of monolayer 1T-TaSe$_2$[20] and bulk 1T-TaS$_2$[31]. For the LHB energy, the STS maps of all the three systems display the same patterns where the electrons concentrating at the center of SOD motifs. In contrast, for the UHB energy, the STS maps of bulk 1T-TaS$_2$ exhibit similar patterns to those of LHB, while they exhibit completely different patterns of monolayer 1T-NbSe$_2$ and monolayer 1T-TaSe$_2$ where electrons prefer to locate around the outer rim of SOD motifs.

Previously, it has been theoretically demonstrated that the interaction between the localized and itinerant orbitals plays a dominant role in the electronic structures of bulk 1T-TaS$_2$[31]. Our results obviously verify that, on the basis of such a two-orbital model and density functional theory (DFT) + U simulations[32,33], additional Coulomb interactions should also be taken into account to explain the spatial repulsion of LHB and UHB in monolayer 1T-NbSe$_2$ and monolayer 1T-TaSe$_2$[31]. Although bulk 1T-TaS$_2$ is usually considered to be a quasi-2D system, a slight interlayer interaction may increase electronic delocalization and screening, thus resulting in a reduction of Coulomb interactions[20,31].

The spin fluctuations in monolayer 1T-NbSe$_2$ are further revealed by placing monolayer 1T-NbSe$_2$ in contact with a metallic 1H-NbSe$_2$ (Fig. 1e, f). The STS spectra recorded above the 1T/1H-NbSe$_2$ heterostructure exhibit a pronounced DOS peak at the Fermi level with a FWHM of about 40 meV (Fig. 1g, h), which is observed on all SOD motifs. Moreover, the relative intensities of the zero-bias peaks follow the CDW periodicity of the topmost 1T-NbSe$_2$ and reach the maximum at the top sites. These zero-bias peaks originate from the Kondo resonance[19,26,34] generated by the spin exchange between the local spin in each SOD motif of 1T-NbSe$_2$ and the itinerant electrons in a metallic 1H-NbSe$_2$, as depicted in the inset of Fig. 1g.

### Spinon-modulated carrier density in monolayer 1T-NbSe$_2$

STS maps of monolayer 1T-NbSe$_2$ taken at the same location as the STM image in Fig. 2a and the fast Fourier transforms (FFT) are presented in Fig. 2b–f. For energies away from the Hubbard bands ($E = -1.0$ eV, for example), only the CDW periodicity dominate the FFT image (Fig. 2b, red circles). With energies close to the Hubbard band edges of ±0.2 eV, an additional charge order emerges with the incommensurate lattice constant slightly larger than $\sqrt{3}$ times of the CDW lattice constant (Fig. 2d, f, yellow circles). Such an additional lattice rotates about 30° with respect to the CDW lattice, regardless of the NbSe$_2$-graphene orientation and different STM tips, which help us rule out the influence of graphene substrate and tip-imaging artifacts[35] (Supplementary Note 1 and Supplementary Figs. 6–9). Similar supermodulation behaviors are also acquired at the UHB$_2$ of 0.6 eV (Supplementary Fig. 10), providing significant evidence for strong correlation physics.

One scenario to explain the above observations is to postulate that monolayer 1T-NbSe$_2$ is a U(1) QSL with SFS, which was once reported in an isostructural compound monolayer 1T-TaSe$_2$. In a U(1) QSL, electrons experience a spin-charge separation and decompose into spinons and chargons, both of which couple to a U(1) gauge field. The quantum fluctuations of the U(1) gauge field have effect on the low-energy excitations in the QSL, resulting in a Fermi surface instability that partially gaps the SFS[9,19,36]. Near the Hubbard band edges, the spatially periodic spinon density modulated by the SFS instability is involved in the electron tunneling process and gets

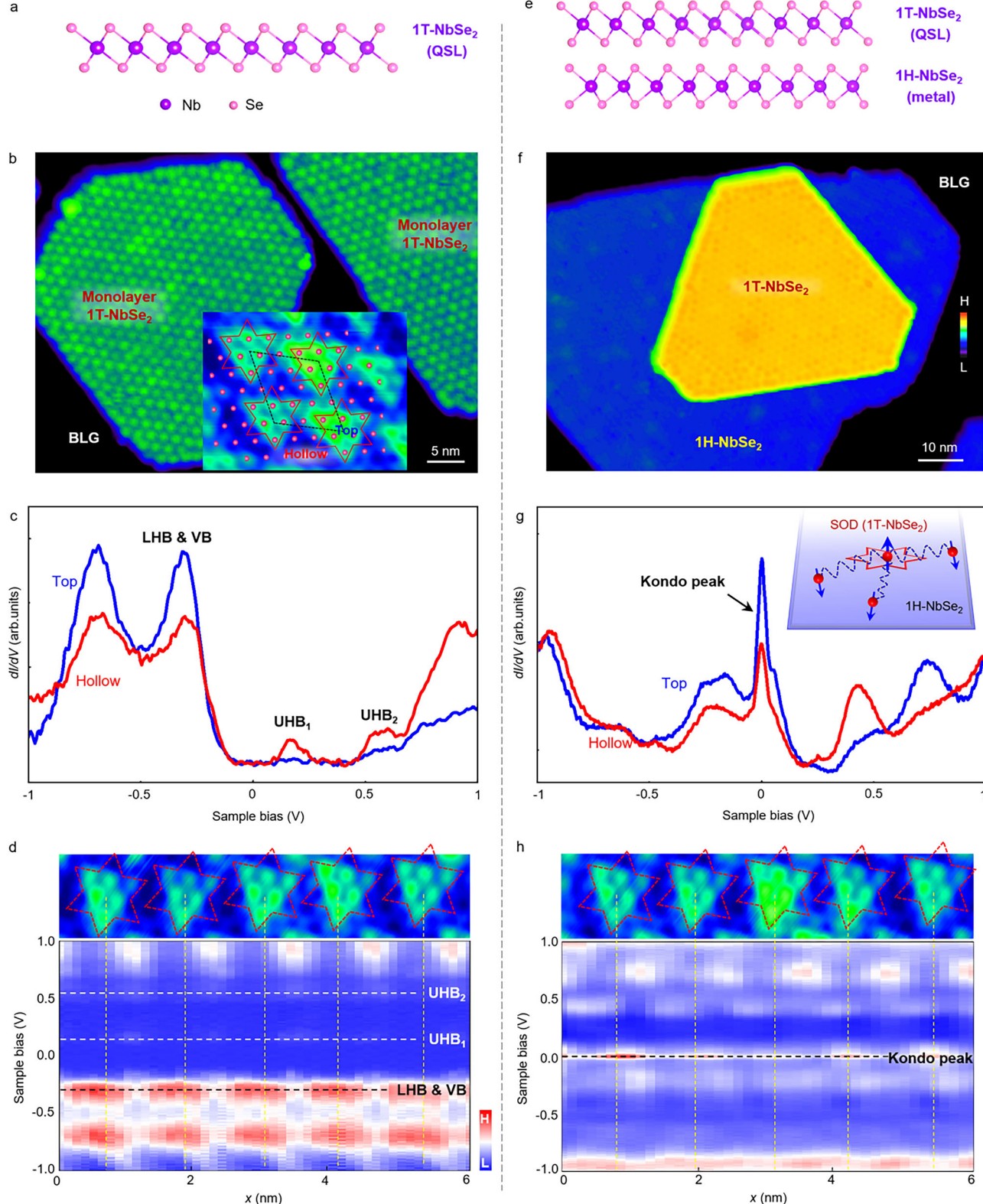

manifested in the STS maps. Therefore, the incommensurate super-modulations reflect the composite density modulations of the itinerant spinons in monolayer 1T-NbSe₂.

**Spinon Kondo effect of MnPc molecules on monolayer 1T-NbSe₂**

The conjecture that 1T-NbSe₂ monolayers comprise a U(1) QSL with SFS gets further supported by the STS spectra of a magnetic molecule adsorbed on top. In our experiment, we sublime a low

dose of MnPc molecules onto monolayer 1T-NbSe₂ (see "Methods"). Figure 3b shows the atomic structure of a MnPc molecule, which is composed of a central Mn ion and the surrounding planar Pc ligand, exhibiting a fourfold symmetry[37]. The out-of-plane $d$-orbitals of the Mn ion, i.e., $d_{z^2}$ and $d_{xz}/d_{yz}$, mainly contribute to the local magnetic moment of $S = 3/2$ spin, as depicted in Fig. 3c[38–40]. Since there is an electronic coupling between the Mn $d$-orbitals and ligand $\pi$-orbitals, the magnetic moment of a MnPc molecule does not localize at the

**Fig. 1 | Correlated insulating states and quantum spin fluctuations in mono-layer 1T-NbSe$_2$. a** Atomic structure of monolayer 1T-NbSe$_2$ in side view. **b** Large-scale STM topographic image of monolayer 1T-NbSe$_2$ on BLG/SiC(0001) substrate ($V_b = -1.5$ V, $I_t = 10$ pA). **c** Typical STS spectra of monolayer 1T-NbSe$_2$ recorded on the top and hollow sites of charge-density-wave (CDW) pattern. The LHB & VB, UHB$_1$, and UHB$_2$ are marked accordingly. Inset: Atomic-resolution STM image of 1T-NbSe$_2$ ($V_b = -1$ V, $I_t = 3$ nA). The top Se atoms dominate the STM image, exhibiting a ($\sqrt{13} \times \sqrt{13}$) $R$13.9° CDW pattern. The bright and dark regions in the image correspond to the sites on and off the SOD, as marked by top and hollow, respectively. **d** Spatially resolved STS spectra recorded along the CDW basis vectors in monolayer 1T-NbSe$_2$, indicating the correlated insulating state. **e** Atomic

structure of 1 T/1H-NbSe$_2$ vertical heterostructure in side view. **f** Large-scale STM topographic image of 1 T/1H-NbSe$_2$ vertical heterostructure on BLG/SiC(0001) substrate ($V_b = -1.5$ V, $I_t = 10$ pA). **g** Typical STS spectra of 1 T/1H-NbSe$_2$ vertical heterostructure recorded on the top and hollow sites of CDW pattern. The sharp peak at the Fermi energy is assigned to a Kondo peak, indicating a considerable interaction between the spin and electron states in the two layers. Inset: Schematic of the Kondo effect. The itinerant electrons in a metal 1H-NbSe$_2$ couple with a local spin in each SOD of 1T-NbSe$_2$, yielding the Kondo screening effect. **h** Spatially resolved STS spectra recorded along the CDW basis vectors in 1 T/1H-NbSe$_2$ vertical heterostructure, reflecting the existence of spin fluctuations in monolayer 1T-NbSe$_2$.

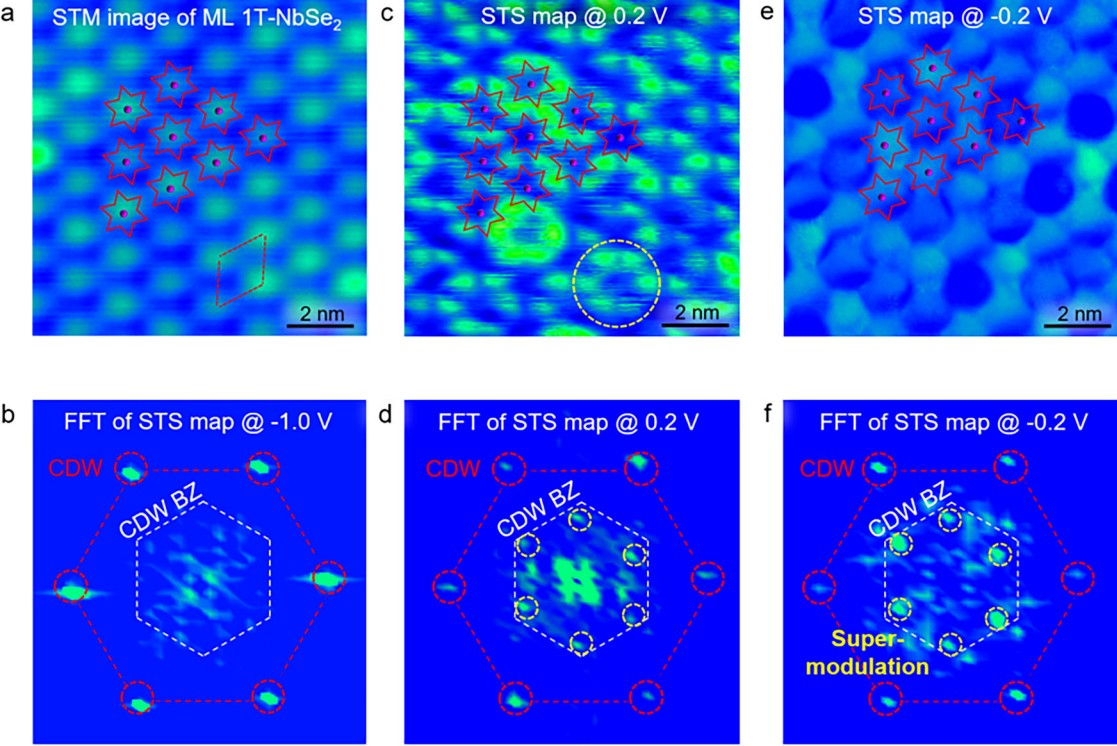

**Fig. 2 | Quantum spin liquids (QSLs) behavior of monolayer 1T-NbSe$_2$. a** STM image of monolayer 1T-NbSe$_2$ recorded at the sample bias of −1 V. **c, e** STS maps of monolayer 1T-NbSe$_2$ recorded at the sample bias of 0.2 V and −0.2 V with the same location as panel (**a**). The CDW lattice and its long-wavelength modulation are marked by the red dots and white circles, respectively. **b, d, f** FFT images of the STS maps at the sample bias of −1.0 V, 0.2 V, and −0.2 V, respectively. The white

hexagon represents the CDW Brillouin zone (BZ). The six bright spots enclosed by the red circles indicate the CDW wavevectors of monolayer 1T-NbSe$_2$, while the spots enclosed by the yellow circles are related to the charge modulation with the wavelength larger than $\sqrt{3}$ times of the CDW wavevectors. This long-range charge modulation originates from spinon excitations.

central Mn ion. Instead, it spatially extends over the whole Pc ligand[41].

Figure 3d, e shows two representative STM images of an individual MnPc molecule on monolayer 1T-NbSe$_2$. The central Mn ion appears as a protrusion when it locates on the SOD motifs of monolayer 1T-NbSe$_2$ (hereinafter simply MnPc-top site, see Fig. 3d) and as a depression when it locates off the SOD motifs (MnPc-hollow site, see Fig. 3e). The two topographic configurations can be reversibly moved via a STM tip manipulation technique (Fig. 3a and Supplementary Fig. 11), implying a weak electronic hybridization between the MnPc molecule and the central Nb $d_{z^2}$-orbitals in each SOD of monolayer 1T-NbSe$_2$. In consideration of the fourfold symmetry of MnPc molecules and the sixfold symmetry of monolayer 1T-NbSe$_2$ with the triangular CDW pattern, the electronic hybridization breaks the fourfold symmetry of the MnPc molecule. In such a case, the four lobes of the MnPc molecules can be roughly classified into Pc-1 and Pc-2, depending on whether the lobe

(not the center molecule) is located on or off a SOD motif, respectively (Fig. 4a, b inset and Supplementary Fig. 11).

Figure 4a, b shows the site-dependent STS spectra of a MnPc molecule on and off a SOD motif in monolayer 1T-NbSe$_2$. The key characteristics is that there are additional low-energy resonance peaks in the STS spectra, as marked by the blue arrows. Specifically, for the MnPc-top, the additional peaks emerge at the band edge in pairs if the STS spectra are acquired at both the central Mn ion and in the configuration denominated Pc-1, while there is no additional peak for the spectra acquired for configurations Pc-2 (Fig. 4a). In contrast, for the MnPc-hollow, the STS spectra acquired at the central Mn ion and Pc-1 usually exhibit one weak additional peak (Fig. 4b). Moreover, the intensities of these additional peaks exhibit a significant spatial inhomogeneity, reaching the maximum at the central Mn ion of the MnPc-top regime (purple line in Fig. 4a, also see Supplementary Note 2, Supplementary Figs. 12 and 13).

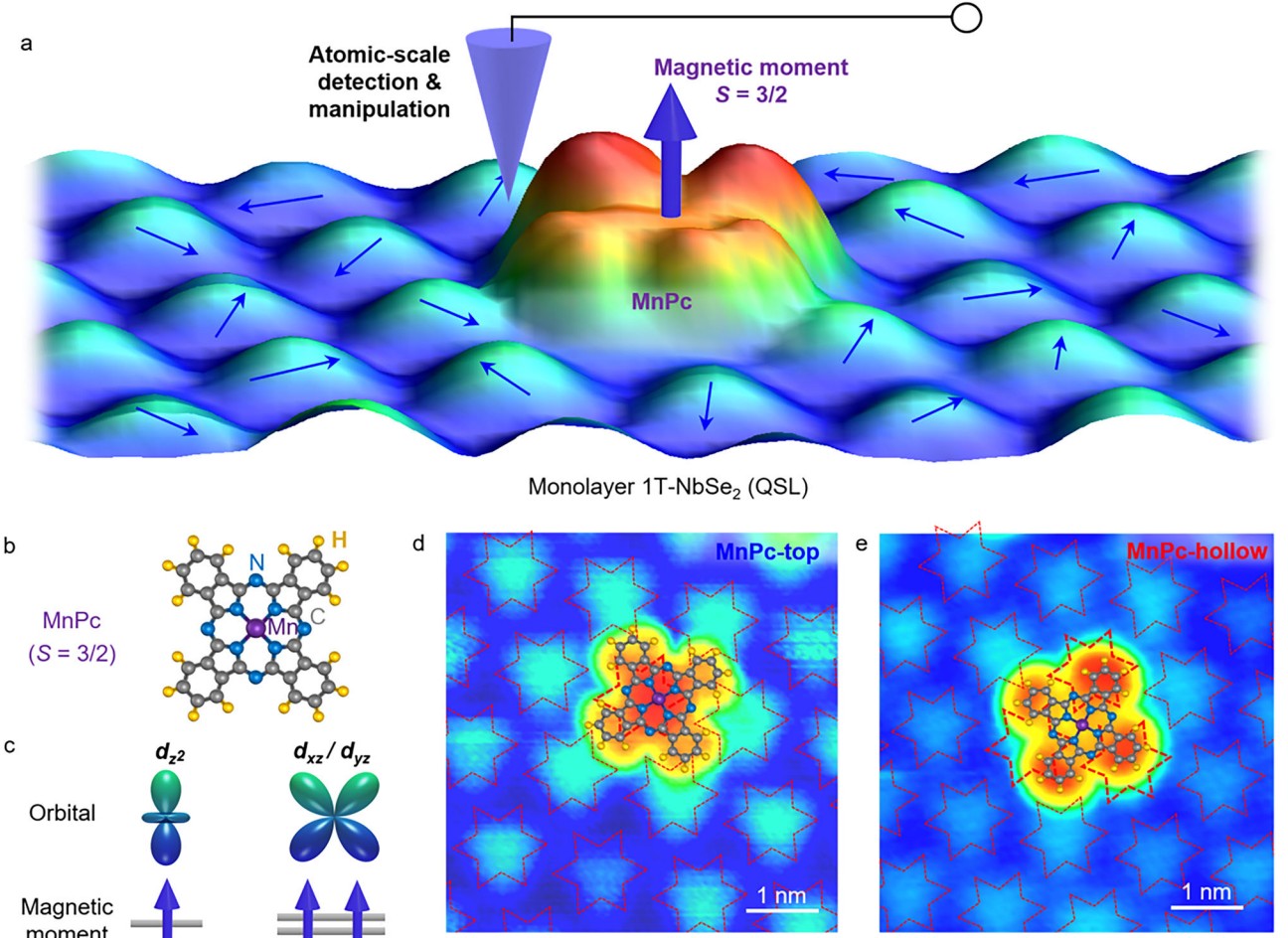

**Fig. 3 | STM topographic signatures of MnPc molecules with spin $S = 3/2$ on monolayer 1T-NbSe$_2$. a** Schematic representation of atomic-scale manipulation and detection technique. The STM tip can reversibly manipulate the MnPc molecule on the top and hollow sites of CDW lattice in monolayer 1T-NbSe$_2$, and can efficiently detect, simultaneously, the interaction between a QSL state (monolayer 1T-NbSe$_2$) and a local magnetic impurity with $S = 3/2$ spin (MnPc). **b** Atomic structure of MnPc molecule, including a center Mn ion and the surrounding Pc ligand. **c** $d$-orbitals and spin configurations of the center Mn ion in MnPc. **d, e** STM images of a MnPc molecule on monolayer 1T-NbSe$_2$, with the central Mn atom of the molecule locating on the top and hollow sites of the CDW lattice, respectively ($V_b = -3.5$ V, $I_t = 10$ pA). The atomic structures of MnPc molecule are superimposed onto the STM images, exhibiting a fourfold symmetry.

For comparison, we also carry out similar STM and STS measurements of a non-magnetic molecule ZnPc on monolayer 1T-NbSe$_2$, as summarized in Supplementary Figs. 14–16. From the site-dependent STS spectra shown in Supplementary Fig. 14, we can find out that there is almost no obvious resonance peaks emerging at the band edges when recorded on the central Zn ion, regardless of the ZnPc molecule locating on or off a SOD motif of monolayer 1T-NbSe$_2$. Our spatially resolved STS maps acquired at the band edge energies further reveal the absence of additional resonance peaks around the non-magnetic ZnPc molecule (Supplementary Figs. 15 and 16), which exhibits significant difference from those of the magnetic MnPc molecule (Supplementary Figs. 12 and 13). These phenomena highlight that the magnetic impurity is a core cause for generating the additional resonance peaks in monolayer 1T-NbSe$_2$.

The pair of resonance peaks at the band edge of monolayer 1T-NbSe$_2$ induced by a magnetic molecule observed in our experiment are in good agreement with the spinon Kondo scenario[27,28] (Fig. 4c, "Methods", Supplementary Note 3, Supplementary Figs. 17–19). Assuming that monolayer 1T-NbSe$_2$ is a U(1) QSL with SFS, the itinerant spinons in the gapless U(1) QSL will form a spinon Kondo screening cloud around a magnetic molecule. For a MnPc molecule with $S = 3/2$ at the underscreened Kondo regime, a remaining local spin with $S = 1$ is "asymptotically" decoupled from the spinon Kondo screening cloud[42].

In the spinon Kondo cloud, due to the spinon-chargon binding that arises from the U(1) gauge field fluctuations in the QSL, a doublon and a holon are attracted to a spinon and a spinon hole respectively, forming the composite spinon-chargon states (Fig. 4d). Given a mild gauge binding interaction $U_R = 0.17$ eV (the theoretical estimation amounts to about half of the spinon band width[28]), the composite spinon-chargon states correspond to an electronic state at the bottom of the UHB and a hole state at the top of the LHB. Importantly, these two states are both effectively bound to the MnPc molecule, as depicted in Fig. 4e. Following this spinon Kondo scenario we idealized the MnPc molecule as a local magnetic impurity and calculated the electronic DOS of band-edge resonance peaks of a QSL by considering different interaction scenarios as shown in Fig. 4c.

In reality, since magnetic MnPc molecules have a finite spatial spread to modify the Coulomb repulsion profile and the spin moment distribution in monolayer 1T-NbSe$_2$, the magnetic coupling between the MnPc magnetic moments and the spins in monolayer 1T-NbSe$_2$ SOD motifs is far more complicated than the idealized local magnetic impurity model we considered above. It is possible that the coupling between the MnPc and the SOD in 1T-NbSe$_2$ changes the charge distribution of the LHB, so the Coulomb repulsion acted on the injected electronic states from the UHB becomes different from that in the pristine case[20]. From our experiments we can find out that the original

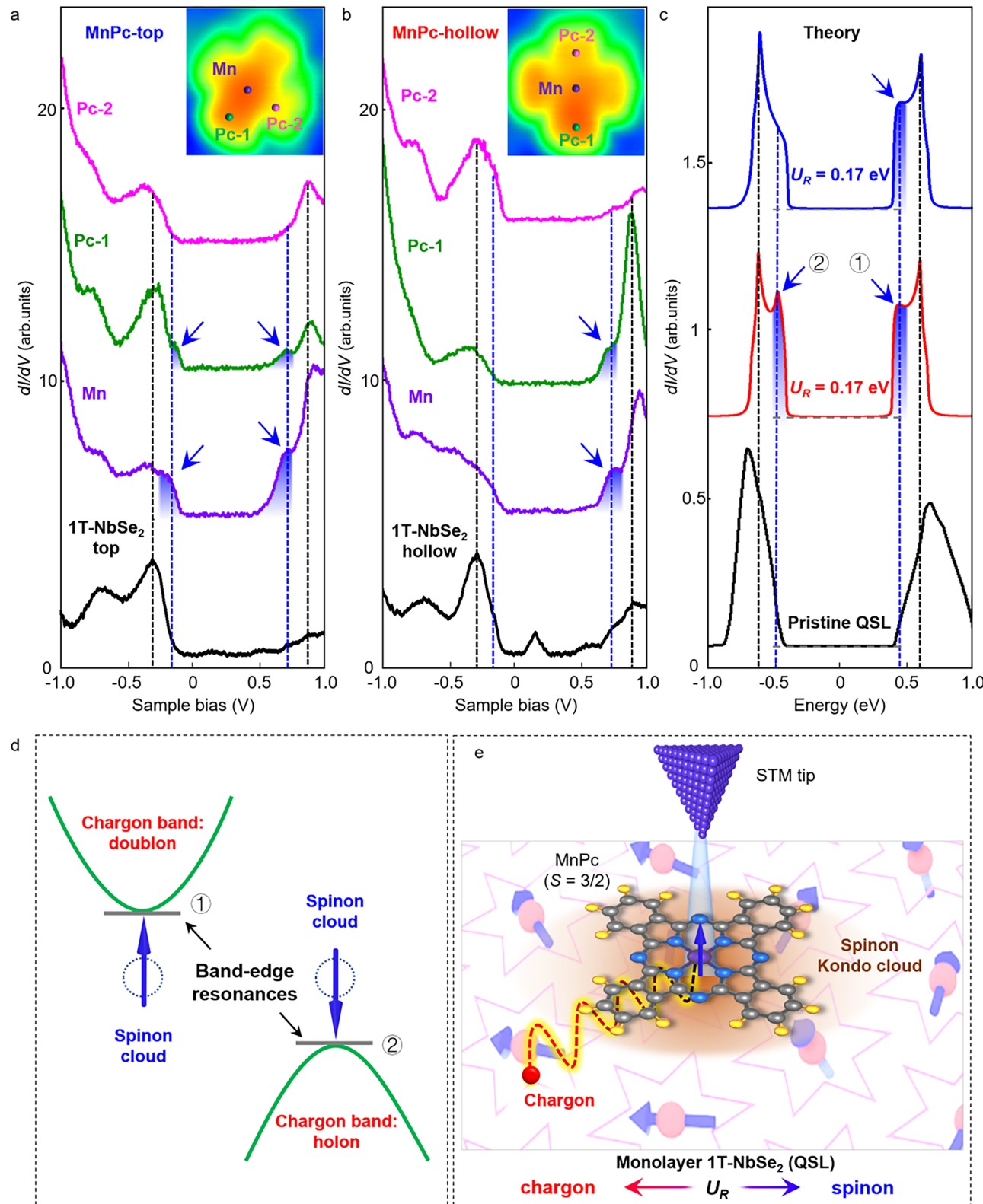

**Fig. 4 | Spinon Kondo effect of MnPc molecules on monolayer 1T-NbSe₂.**
**a**, **b** Site-dependent STS spectra recorded on pristine monolayer 1T-NbSe₂ as well as on MnPc-top and MnPc-hollow, as marked in the inset. The new resonance peaks indicated by arrows appear at the band edges, which arise from the spinon Kondo effect. **c** Theoretical simulation of the electronic DOS that exhibits the emergence of band edge resonance peaks. The DOS in red corresponds to a local magnetic impurity involved in both holon-spinon hole binding and doublon-spinon binding. The DOS in blue corresponds to the case with only doublon-spinon binding. The DOS in black is the electronic DOS of the pristine QSL. The resonance peaks appear at the band edges, with their strength varying with the spin-charge binding inter-action. **d** Energy diagram of the band edge resonance states. **e** Physical mechanism of the interaction between a QSL state and a local magnetic moment. A magnetic impurity in a QSL candidate can result in a spinon Kondo screening cloud (brown shadow). Such a cloud attracts a chargon in QSL under the spin-charge binding interaction.

UHB$_1$ and UHB$_2$ undergo a spectral weight re-distribution under the MnPc-modified Coulomb potential profile, resulting in the UHB$_1$ suppressed and the UHB$_2$ merging into a higher energy. Although we cannot quantitatively explain such a phenomenon, we believe UHB states are quite sensitive to the local environment because atomic defects in monolayer 1T-NbSe$_2$ can also influence the energies and intensities of the UHB$_1$ and UHB$_2$ in a similar way (Supplementary Fig. 20). We can see from Fig. 4, both the magnetic moment distribution inside a MnPc molecule and the relative position of the MnPc molecule and the SOD motif below have significant impact on the spectral features at the Hubbard band edges. For the spectra taken in the "Mn" and "Pc-1" positions of the "MnPc-top" configuration (purple and green curves in Fig. 4a), there are a pair of resonance peaks emerging at the Hubbard band edges, which can be best captured by the idealized local magnetic impurity model. For the spectra taken from the "Pc-2" lobe of MnPc (pink curve in Fig. 4a) the magnetic moment of the MnPc molecule may couple differently such that the spin moment is less distributed across the molecule, leading to absence of the expected band edge resonance peaks. For the MnPc molecule deposited off the SOD motif, the small overlap between the magnetic moment in the MnPc molecule and the holon state in the QSL monolayer 1T-NbSe$_2$ can effectively reduce the binding of a holon and a spinon hole, so the band edge resonance peak near the LHB is absent, leaving only one resonance peak in the electronic DOS spectra (purple and green curves in Fig. 4b and blue curve in Fig. 4c).

However, for a non-magnetic ZnPc molecule with $S = 0$ spin, there is no spin exchange between the ZnPc molecule and the itinerant spinons, since a ZnPc molecule hosts no local spin. As a result, the additional resonance peaks are not expected to appear at the band edge of monolayer 1T-NbSe$_2$, which is in well consistent with our measurements shown in Supplementary Figs. 14–16. Therefore, the observed site-dependency of the resonance peaks generated by a magnetic MnPc molecule, complementary by the absence of resonance peaks generated by a non-magnetic MnPc molecule, are unambiguously demonstrate that the additional resonance peaks at the band edges are attributed to the spinon Kondo scenario of a magnetic impurity deposited on a U(1) QSL with SFS.

## Discussion

Since the QSL states were first predicted in 1973, many theoretical predictions have been focused on the QSL models of 2D triangular-lattice correlated insulators. Experimentally, several single-crystal and organic materials with triangular atomic or molecular layers have been demonstrated to be QSL candidates[10,11,43,44]. Until 2021, monolayer 1T-TaSe$_2$ was verified to be the first QSL candidate in the 2D limit[19]. In this work, we provide the experimental evidence of the QSL signatures in monolayer 1T-NbSe$_2$, another 2D system exhibiting an isostructural compound with monolayer 1T-TaSe$_2$. Indeed, most of our observations in monolayer 1T-NbSe$_2$ are similar as those reported in monolayer 1T-TaSe$_2$. Even so, the QSL behavior in monolayer 1T-NbSe$_2$ should not be directly viewed as a simple extrapolation from 1T-TaSe$_2$ to 1T-NbSe$_2$ by intuition, but needs to be carefully investigated, because there are still many differences in electronic properties[10,19,20,27,45,46], as summarized in Supplementary Note 4 and Supplementary Table 1.

Our experimental phenomena can further help us rule out the possibility of other QSL states in monolayer 1T-NbSe$_2$, such as the Z$_2$ QSL state and the Dirac QSL. Although Z$_2$ QSL state was suggested in the isostructural bulk 1T-TaS$_2$[10,11], it is unlikely to be the ground state of monolayer 1T-NbSe$_2$. One essential reason is that the Z$_2$ QSL has a fully gapped spinon spectrum, any spin excitation in Z$_2$ QSL requiring an excitation energy to overcome the spinon gap. In such a case, the Kondo resonance should not occur at the Fermi energy as a Z$_2$ QSL contacting with a metal. Moreover, a fully gapped Z$_2$ QSL indicates that there are no itinerant spinons in the ground state, so no spinon Kondo effect is expected. In our experiment, the Kondo resonance peak at the

Fermi energy can be clearly observed by placing monolayer 1T-NbSe$_2$ onto a metallic 1H-NbSe$_2$, and the spinon Kondo effect can be observed by depositing a magnetic impurity onto monolayer 1T-NbSe$_2$. Based on our experimental observations, we can definitely rule out the possibility of the gapped Z$_2$ QSL state.

Besides the U(1) QSL, the Dirac QSL is also allowed to exhibit gapless spin excitations[2–4,47,48]. However, previous predictions proposed that an external gauge field flux is usually required to stabilize the Dirac QSL[2–4,47,48]. Since our measurements are all carried out in the absence of magnetic field, the gapless spin excitations indicated by our observations in monolayer 1T-NbSe$_2$ are more likely from the SFS rather than the Dirac-type spinon spectrum. Therefore, our results strongly suggest that monolayer 1T-NbSe$_2$ is a U(1) QSL with SFS.

Our proposed ground state of QSL with SFS is also stable against weak disorders[49]. In the slave rotor mean field picture[50], the random distributed impurities will induce disorder potentials in the spinon channel. In the weak disorder regime where the impurity potential range is smaller than the correlated gap size[49], the spinon disorders are expected to play the similar role as that of the ordinary disorders in a metal[13]. In our monolayer 1T-NbSe$_2$ samples, there is only a low concentration of atomic defects, the energy potential of which cannot be comparable to the correlated gap size, indicating that our samples are in the weak disorder regime. Since the ground state of SFS is protected by the correlated gap, the weak disorder potentials may not affect the nature of the emerging QSL proposed in our samples.

In summary, our STM/STS measurements reveal the signatures of the gapless U(1) QSL phase in monolayer 1T-NbSe$_2$. Firstly, by supporting monolayer 1T-NbSe$_2$ on a metallic 1H-NbSe$_2$, we observe periodic Kondo resonance peaks. This phenomenon confirms the presence of a local spin in each SOD motif across the CDW patterns, and also that the spin-orientation shows free quantum fluctuation. Secondly, the STS maps reveal a long-wavelength charge density modulation when recorded at the Hubbard band energies, which is attributed to spinon excitations with a partially gapped SFS. Thirdly, by further depositing magnetic MnPc molecules with $S = 3/2$ onto monolayer 1T-NbSe$_2$, there are additional resonance peaks emerging at the Hubbard band edges of monolayer 1T-NbSe$_2$. These are consistent with the spinon Kondo effect of a magnetic impurity deposited on a U(1) QSL with SFS. Our experimental observations sequentially demonstrate that monolayer 1T-NbSe$_2$ hosts local spin excitations, contains correlated in-gap carriers, and readily causes a Kondo screening cloud formed by composite spinon-chargon states around a deposited magnetic impurity. Taken together, these results strongly indicate that monolayer 1T-NbSe$_2$ is a U(1) QSL with partially gapped SFS.

As a new platform that hosts a QSL state, monolayer 1T-NbSe$_2$ allows for further investigations on the QSL physics in 2D quantum materials, particularly including the in-gap exotic states and the QSL-superconductivity interaction in van der Waals heterostructures[17,51–53]. Moreover, the collective nature of QSL in triangular lattice with SFS ground state can be further probed, as theoretically predicted very recently[54,55].

## Methods

### Sample preparation

The bilayer graphene (BLG) was obtained by thermal decomposition of 4H-SiC(0001) at 1220 °C for 40 min. NbSe$_2$ layers were epitaxially grown on BLG/SiC(0001) by evaporating Se and Nb from an electron beam evaporator and a Knudsen cell evaporator, respectively. The flux ratio of Se and Nb is more than 10:1 to guarantee a Se-rich environment. The BLG/SiC(0001) substrate was maintained at 500 °C during the growth, followed by a post-annealing process at 400 °C for 20 min. MnPc molecules (Sigma-Aldrich) were first purified via a vacuum sublimation, and then were thermally deposited from a Knudsen cell evaporator to NbSe$_2$/BLG/SiC(0001) at 345 °C for 30 min.

## STM/STS measurements

STM/STS measurements were performed using a custom-designed STM system at 4.2 K under ultrahigh-vacuum conditions (USM-1300, Unisoku). An electrochemically etched tungsten tip was used as the STM probe, which was calibrated by using a standard graphene lattice and a Si(111)-(7 × 7) lattice. The STS measurements were taken by a standard lock-in technique with the bias modulation of 2 mV at 973 Hz.

## Theoretical calculations

In the electronic DOS calculation, we take the slave rotor mean field approximation to describe the U(1) QSL with SFS. In the QSL, electrons are fractionalized into spinons and chargons. The spinon and chargon band dispersions are

$$
\xi_{\boldsymbol{k}} = 2t_{f1}\left(2\cos\frac{1}{2}k_x a\cos\frac{\sqrt{3}}{2}k_y a + \cos k_x a\right) \\
+ 2t_{f2}\left(2\cos\frac{3}{2}k_x a\cos\frac{\sqrt{3}}{2}k_y a + \cos\sqrt{3}k_y a\right) - \mu_f \tag{1}
$$

$$
\epsilon_{\boldsymbol{k}} = 2t_{X1}\left(2\cos\frac{1}{2}k_x a\cos\frac{\sqrt{3}}{2}k_y a + \cos k_x a - 3\right) \\
+ 2t_{X2}\left(2\cos\frac{3}{2}k_x a\cos\frac{\sqrt{3}}{2}k_y a + \cos\sqrt{3}k_y a\right) + \Delta \tag{2}
$$

where $\boldsymbol{k} = \left(k_x, k_y\right)$ denotes the quasi-momentum of the Bloch states in a triangular lattice and $a$ is the lattice constant. Here $\Delta = 0.4$ eV is fixed by the experimentally observed correlated insulating gap size. The hopping parameters are set to be $t_{f1} = 0.06$ eV (spinon coupling between the nearest-neighbor SOD motifs), $t_{f2} = -0.006$ eV, $t_{X1} = -0.03$ eV and $t_{X2} = 0.003$ eV to have the best match with the experimentally observed DOS spectrum. The chemical potential $\mu_f = -0.038$ eV is determined by the half-filling requirement of the spinon band.

For the $S = 3/2$ magnetic impurity that is in the underscreening Kondo regime, there remains $S = 1$ spin decoupled in the impurity while $S = 1/2$ spin couples with the nearby itinerant spin-1/2 particles[42]. We focus on the $S = 1/2$ spin channel that is involved in the Kondo coupling. We know that the local magnetic impurity deposited on the U(1) QSL with SFS couples with the spinons and chargons in the QSL. The Matsubara Green's function that involves the coupling in spinon channel takes the form $G_{fs}^{-1}(i\omega_n) = \begin{pmatrix} i\omega_n - \epsilon_0 + h & -w \\ -w & G_{f,\sigma}^{-1}(i\omega_n, \boldsymbol{R}, \boldsymbol{R}) \end{pmatrix}^{-1}$ with $G_{f,\sigma}(i\omega_n, \boldsymbol{R}, \boldsymbol{R}) = \sum_{\boldsymbol{k}} \frac{1}{i\omega_n - \xi_{\boldsymbol{k}}}$ The Matsubara Green's function that involves the coupling in the chargon channel is $G_{bc}^{-1}(i\nu_n) = \begin{pmatrix} G_{dh}^{-1}(i\nu_n) & \hat{u} \\ \hat{u} & G_{DH}^{-1}(i\nu_n, \boldsymbol{R}, \boldsymbol{R}) \end{pmatrix}^{-1}$ with $G_{dh}(i\nu_n) = \begin{pmatrix} \frac{1}{i\nu_n - \sqrt{\lambda + \frac{h^2}{U}} + h} & 0 \\ 0 & -\frac{1}{i\nu_n - \sqrt{\lambda + \frac{h^2}{U}} + h} \end{pmatrix}$, $G_{DH}(i\nu_n, \boldsymbol{R}, \boldsymbol{R}) = \sum_{\boldsymbol{k}} \begin{pmatrix} \frac{1}{i\nu_n - \epsilon_{\boldsymbol{k}}} & 0 \\ 0 & \frac{-1}{i\nu_n + \epsilon_{\boldsymbol{k}}} \end{pmatrix}$, and $\hat{u} = u\sqrt{\frac{U}{2\sqrt{h^2 + \lambda U}}}\begin{pmatrix} 1 & 1 \\ 1 & 1 \end{pmatrix}$. Here $h, \lambda, w, u$ are the mean field parameters that describe the coupling between the local magnetic impurity and the QSL. $\epsilon_0$ is the onsite energy of the local state at the magnetic impurity. $U$ is the Coulomb repulsion in the magnetic impurity. All these parameters are given in the Supplementary Information. For integrating the spinon channel and the chargon channel into the electronic channel, the Matsubara Green's function for the electronic states can be obtained through the convolution

$$
G_{0,\sigma}(i\omega_n) = -\frac{1}{\beta}\sum_{\nu_n}\begin{pmatrix} G_{fs}^{11}(i\omega_n + i\nu_n)G_{bc}^{11}(i\nu_n) & G_{fs}^{12}(i\omega_n + i\nu_n)G_{bc}^{12}(i\nu_n) \\ G_{fs}^{21}(i\omega_n + i\nu_n)G_{bc}^{21}(i\nu_n) & G_{fs}^{22}(i\omega_n + i\nu_n)G_{bc}^{22}(i\nu_n) \end{pmatrix} \tag{3}
$$

Here $G_{0,\sigma}(i\omega_n)$ is a $4 \times 4$ matrix, $G_{fs}^{ij}(i\omega_n)$ denotes the $(i, j)$th matrix element in $G_{fs}(i\omega_n)$, and $G_{bc}^{ij}(i\nu_n)$ denotes the $(i, j)$th matrix block of $2 \times 2$ in $G_{bc}(i\nu_n)$. The spinon-chargon binding induced by the U(1) gauge field fluctuations in the QSL will further modify the electronic Matsubara Green's function to be $G_\sigma(i\omega_n) = \left[1 - G_{0,\sigma}(i\omega_n)\hat{U}\right]^{-1}G_{0,\sigma}(i\omega_n)$ with $\hat{U} = \begin{pmatrix} 0 & 0 \\ 0 & \sigma_z \end{pmatrix}$.

Here $\sigma_z$ is the Pauli matrix. The Matsubara Green's function for the electronic state at the impurity then reads

$$
G_{d,\sigma}(i\omega_n) \\
= \sum_{i,j=1,2}\left\{G_{0,\sigma}^{11}(i\omega_n) + G_{0,\sigma}^{12}(i\omega_n)U_b\sigma_z\left[1 - G_{0,\sigma}^{22}(i\omega_n)U_b\sigma_z\right]^{-1}G_{0,\sigma}^{21}(i\omega_n)\right\}_{ij} \tag{4}
$$

After analytic continuation $i\omega_n \to \omega + i0^+$, the corresponding retarded Green's function is obtained as $G_{d,\sigma}^R(\omega) = G_{d,\sigma}(i\omega_n \to \omega + i0^+)$. The local electronic DOS of the magnetic impurity deposited on the QSL is then $\rho_{d,\sigma}(\omega) = -\frac{1}{\pi}\mathrm{Im}G_{d,\sigma}^R(\omega)$, which gives the blue line in Fig. 4c.

## Data availability

The data generated in this study are available within the article and its Supplementary Information files or from the corresponding author upon request. Source data are provided as a Source data file. Source data are provided with this paper.

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

## Acknowledgements

Y.Z. acknowledges the National Key R&D Program of China (Nos. 2022YFA1402602, 2022YFA1402502), National Natural Science Foundation of China (No. 12274026), and China Postdoctoral Science Foundation (No. 2021M700407). Y.L.W. thanks the National Key R&D Program of China (Nos. 2021YFA1400103, 2020YFA0308802), National Natural Science Foundation of China (Nos. 92163206, 12321004). Q.Z.Z. acknowledges the National Natural Science Foundation of China (No. 62101037). H.X.Y. acknowledges the National Natural Science Foundation of China (No. 12304205). L.W.L. acknowledges the National Natural Science Foundation of China (No. 62371041). T.A.J. is grateful for support by the Swiss Nanoscience Institute and the Swiss National Science Foundation (No. 200020_207769 and predecessor projects). W.Y.H. acknowledges the support from National Natural Science Foundation of China (No. 12304200), the BHYJRC Program from the Ministry of Education of China (No. SPST-RC-10), and the start-up funding from ShanghaiTech University.

## Author contributions

Y.Z., H.J.G., and Y.L.W. coordinated the research project. Q.Z.Z., Y.Y.C., L.G.J., Y.H.H., H.Y.J., H.X.Y., T.Z., and L.W.L. synthesized the samples and performed the STM experiments. W.Y.H. performed the theoretical calculations. Y.Z., W.Y.H., T.A.J., and Y.L.W. contributed to the overall scientific interpretation and edited the manuscript. All authors were involved in discussions of this work.

## Competing interests

The authors declare no competing interests.
