## [Peer Review File · Nature Communications]

REVIEWER COMMENTS

Reviewer #1 (Remarks to the Author):

The paper »Quantum spin liquid signatures in monolayer 1T-NbSe2« submitted by Quanzhen Zhang et al. presents experimental observation and theoretical modelling of QSL signatures in monolayer 1T-NbSe2.

The relevance of this work: Realisations of QSL are, despite ongoing research efforts, still surprisingly scarce. Few years ago, Patrick Lee suggested that 1T-TaS2 might be an intriguing candidate for QSL and this has triggered several theoretical and experimental activities. Present submission extends the idea to the related 1T-NbSe2, where, just like in 1T-TaS2, QSL is supposed to develop within the CDW phase. In principle, QSL in these systems could be either fully gapped Z2 spin liquid or gapless Dirac spin liquid. In this work authors claim to have evidences for the U(1) QSL with SFS. This is important as it narrows down possible QSL. Therefore, I conclude that this submission could be potentially very interesting for the QSL community, but could attract also the general readership of Nature Communications.

Quality of research: Authors in this work mainly rely on the low-temperature STM techniques. First, they provide evidence for the presence of a local spin in each SOD motif through the observation of periodic Kondo resonance peaks. Second, they do STS to detect spinon excitations, which are a hallmark of QSL. Finally, they deposit magnetic MnPc molecules to detect spinon Kondo effect of a magnetic impurity. The methodology and the data are of very high quality and certainly meet the high standards of Nature Communications.

Despite my positive assessment of this work, I still think that authors should try to improve their work by considering the following comments:

(1) In a detailed STM study of 1T-TaS2 published in PHYSICAL REVIEW X 7, 041054 (2017) authors identified two distinct types of orbital textures — one localized and the other extended—and demonstrated that the interplay between them is the key factor that determines the electronic structure. Can authors comment on these suggestions and how they are related to their 1T-NbSe2 monolayer?

(2) The conclusion on U(1) QSL is important. In general, the gapless spin liquids generically have fermionic spinons, which may form a Fermi surface or Dirac nodes. This point has not been discussed in their submission. Can authors discuss it? For the QSL in 1T-TaS2 F. Pratt et al suggested Z2 QSL phases (npj Quantum Materials volume 6, 69 (2021)), so a different scenario as claimed in the present submission. Can authors definitely rule out this possibility?

(3) The spinon Fermi surface is characterized by a hopping matrix element of order J . What is the order of magnitude for J between these SOD localised spins in 1T-NbSe2 sample?

(4) Authors avoided completely the discussion of the role of disorder. How important is for the emerging QSL in their sample?

Minor comments:

(1) I think authors may have overlooked some important experimental papers that previously claimed QSL in 1T-TaS₂. Beside the above mentioned PR X 7, 041054 (2017) and npj Quantum Materials volume 6, 69 (2021), I can also think of Nature Physics 13, 1130–1134 (2017).

(2) Is the fact that they are working on ML and not on a bulk sample relevant? Do they expect similar physics also for the bulk samples?

(3) Are the quantum fluctuations of the U(1) gauge field the only possible reason for the opening of a gap in the spinon spectrum?

Reviewer #2 (Remarks to the Author):

The authors study monolayer 1H- and 1T-NbSe₂ films using scanning tunneling microscopy and spectroscopy. The experimental data is solid; They observe (1) 3 x 3 CDW on monolayer 1H-NbSe₂ films, (2) the star-of-David pattern with the Mott insulating gap on monolayer 1T-NbSe₂ films, (3) the Kondo peak on 1T/1H-NbSe₂ heterostructures, (4) incommensurate density waves on monolayer 1T-NbSe₂ films, and (5) resonance peaks at the Mott gap edge at magnetic impurities on monolayer 1T-NbSe₂ films.

Before recommending publication, I hope the authors clarify the advances they made in this study.

Among the observations, all (1)-(5) are already known for TaS₂ and TaSe₂ (ref. 14,17,18,25), and (1)-(3) are known for NbSe₂ (ref. 22-24). Given these similarities, (4) and (5) are also highly likely to be observed for NbSe₂. In other words, from these similarities, NbSe₂ is already expected to be a candidate of U(1) QSL if TaS₂ and TaSe₂ are so. Under this circumstance, the authors actually observe (4) and (5) and interpret them in the same manner as TaS₂ and TaSe₂. I understand that (4) and (5) have never been observed in NbSe₂. Meanwhile, what do we learn from the confirmation of expected behaviors?

Reviewer #3 (Remarks to the Author):

The authors prepared a monolayer 1T-NbSe₂ film by MBE and investigated its electronic structures by STM/STS. 1T-NbSe₂ is naturally expected to possess similar electronic structures to its sister compounds 1T-TaSe₂ and 1T-TaS₂, well-known candidates for the quantum spin liquid. The authors observed a Kondo-like resonance in monolayer 1T-NbSe₂ placed on metallic 1H-NbSe₂, which suggests the local spin at the Star-of-David CDW cluster. They also observed the incommensurate spatial modulations in the electronic state of 1T-NbSe₂ on graphene, which may be related to the spinon Fermi surface in the quantum spin liquid. In addition, the authors observed a pair of resonance peaks in the tunneling spectra taken near the MnPc molecule deposited on 1T-NbSe₂. They state that the observed peaks can be explained by the model based on the quantum spin liquid. Overall, the authors claim that 1T-NbSe₂ is a new promising quantum spin liquid material.

The Kondo-like resonance and the incommensurate electronic state modulations have already been observed in 1T-TaSe₂, and the discussion given in this manuscript basically follows that of the previous publications. The effect of the deposited MnPc molecule is new and interesting. However, I think that the reported results are still premature. I recommend that the authors shorten the discussion on the Kondo resonance and the incommensurate modulation and perform more systematic experiments and theoretical analyses on the effects of magnetic molecules. More specific comments are listed below.

1. I recommend that the authors explain why the UHB splits into UHB1 and UHB2.
2. In Fig. 1, each Star-of-David CDW cluster is imaged as a triangular object, which apparently breaks inversion symmetry. Please explain the reason for this.
3. The authors mention that they can rule out the influences of the graphene substrate and the tip effect as the origin of the observed incommensurate electronic-state modulations. I recommend that the authors show the corresponding experimental data in the supplementary information.
4. In Fig. 4a and 4b, the MnPc-induced peaks are subtle, and it is unclear for me to recognize the site dependence argued in the text. I recommend that the authors quantify the peak intensities and show their spatial variations in 2D images to demonstrate the site dependence clearly.
5. The band-edge resonances due to the spinon Kondo effect occur near the top of the holon band and the bottom of the doublon band. If I understand correctly, the holon and the doublon bands are essentially LHB and UHB, respectively. In the experiment, one peak indeed appears near the top of

the LHB, but another peak is observed near 0.7 eV, which is higher than both UHB1 and UHB2 in the pristine regime. Does the MnPc modify the Mott gap to produce a single UHB at higher energy? If so, please explain the mechanism in detail.

6. I recommend that the authors experimentally and theoretically investigate how the band-edge resonances change when other metal phthalocyanine molecules are deposited. This experiment should be technically simple and should provide information about the spinon Kondo effect for different spins.

Reply to Reviewers' comments and suggestions

We thank all the three Reviewers for their comments and suggestions for a better presentation of the paper. We have carefully considered all of the comments. In the following, we reply to the Reviewers' comments and suggestions point by point.

Reply to the first Reviewer's report

The paper »Quantum spin liquid signatures in monolayer 1T-NbSe₂« submitted by Quanzhen Zhang et al. presents experimental observation and theoretical modelling of QSL signatures in monolayer 1T-NbSe₂.

The relevance of this work: Realisations of QSL are, despite ongoing research efforts, still surprisingly scarce. Few years ago, Patrick Lee suggested that 1T-TaS₂ might be an intriguing candidate for QSL and this has triggered several theoretical and experimental activities. Present submission extends the idea to the related 1T-NbSe₂, where, just like in 1T-TaS₂, QSL is supposed to develop within the CDW phase. In principle, QSL in these systems could be either fully gapped Z₂ spin liquid or gapless Dirac spin liquid. In this work authors claim to have evidences for the U(1) QSL with SFS. This is important as it narrows down possible QSL. Therefore, I conclude that this submission could be potentially very interesting for the QSL community, but could attract also the general readership of Nature Communications.

Quality of research: Authors in this work mainly rely on the low-temperature STM techniques. First, they provide evidence for the presence of a local spin in each SOD motif through the observation of periodic Kondo resonance peaks. Second, they do STS to detect spinon excitations, which are a hallmark of QSL. Finally, they deposit magnetic MnPc molecules to detect spinon Kondo effect of a magnetic impurity. The methodology and the data are of very high quality and certainly meet the high standards of Nature Communications.

Despite my positive assessment of this work, I still think that authors should try to improve their work by considering the following comments.

Reply: We thank the Reviewer for the positive comment “*I conclude that this submission could be potentially very interesting for the QSL community, but could attract also the general readership of Nature Communications*” and “*The methodology and the data are of very high quality and certainly meet the high standards of Nature Communications*”. We have carefully revised our manuscript according to the Reviewer’s valuable suggestions and comments. We hope our response below can satisfactorily address all the comments.

(1) In a detailed STM study of 1T-TaS₂ published in PHYSICAL REVIEW X 7, 041054 (2017) authors identified two distinct types of orbital textures - one localized and the other extended - and demonstrated that the interplay between them is the key factor that determines the electronic structure. Can authors comment on these suggestions and how they are related to their 1T-NbSe₂ monolayer?

Reply: We thank the Reviewer for the valuable comment and the helpful reference. The model proposed in Ref. [*Phys. Rev. X* 7, 041054 (2017)] emphasizes the significance of the interplay between the localized and extended orbital textures on the electronic structure of bulk 1T-TaS₂. We agree with this model that the orbital localized to the central transition metal atom in each SOD motif should be responsible for the Mott transition. This is because that the inter-cluster hopping of the central orbital is a high-order process, so its kinetic hopping is effectively suppressed, and the electronic state becomes susceptible to the electronic repulsion that causes the Mott transition. Indeed, this model can explain most of our experimental phenomena in monolayer 1T-NbSe₂, such as the correlated insulating behavior and the LHB electrons concentrating at the center of SOD motifs. However, additional electron-electron interactions should be considered to capture the UHB electronic behavior in monolayer 1T-NbSe₂, owing to the reduced

screening in two dimensions. More details are given in the following.

Figure R1 summarizes the mean experimental phenomena of the orbital textures measured in bulk 1T-TaS₂ [*Phys. Rev. X* 7, 041054 (2017)], monolayer 1T-TaSe₂ [*Nat. Phys.* 16, 218 (2020) and *Nat. Phys.* 17, 1154 (2021)], and monolayer 1T-NbSe₂ (this work), respectively. The dI/dV maps measured at LHB for all the three systems display similar orbital textures where the electrons concentrating at the center of SOD motifs. However, the spatial distribution of the UHB electrons for the three systems are quite different. In bulk 1T-TaS₂, the dI/dV maps at UHB exhibit a similar pattern as that of LHB (top panel in Fig. R1). In contrast, the dI/dV maps at UHB of monolayer 1T-TaSe₂ and monolayer 1T-NbSe₂ exhibit completely different orbital texture where electrons prefer to locate around the outer rim of the SOD motifs (middle and bottom panels in Fig. R1).

The differences of UHB electronic distributions for the three systems can be well explained by the Coulomb interactions. During the process of electrons injecting from a STM tip into monolayer 1T-TaSe₂ or 1T-NbSe₂, the LHB preferentially occupies the orbital at the center of SOD, thus creating a Gaussian Coulomb repulsion. Such a Coulomb repulsion spatially repulsive to the UHB, in response to the enhanced Coulomb interactions that arise from reduced screening in 2D systems. Although bulk 1T-TaS₂ is usually described as a quasi 2D system, the slight interlayer interaction may increase electronic delocalization and screening, thus resulting in a reduction of Coulomb interactions.

In our revised manuscript, the measured orbital textures of LHB & VB, UHB₁ and UHB₂ in monolayer 1T-NbSe₂ have been added in Supplementary Fig. 5. The Ref. [*Phys. Rev. X* 7, 041054 (2017)], as well as several sentences have been added on Pages 5 and 6 to compare our results with the model proposed in Ref. [*Phys. Rev. X* 7, 041054 (2017)], according to the Reviewer's valuable comment.

Fig. R1. Summarization of the energy-dependent charge distribution in bulk 1T-TaS₂ (extracted from Ref. [*Phys. Rev. X* 7, 041054 (2017)]), monolayer 1T-TaSe₂ (extracted from Refs. [*Nat. Phys.* 16, 218 (2020)] and [*Nat. Phys.* 17, 1154 (2021)]), and monolayer 1T-NbSe₂ (this work).

- (2) The conclusion on U(1) QSL is important. In general, the gapless spin liquids generically have fermionic spinons, which may form a Fermi surface or Dirac nodes. This point has not been discussed in their submission. Can authors discuss it? For the QSL in 1T-TaS₂ F. Pratt et al suggested Z₂ QSL phases (npj Quantum Materials

volume 6, 69 (2021)), so a different scenario as claimed in the present submission. Can authors definitely rule out this possibility?

Reply: We thank the Reviewer for the valuable comment. As the Reviewer said, in general, there are two kinds of gapless spin liquids: one is the spin liquid with spinon Fermi surface and the other is the Dirac spin liquid. For the Dirac spin liquid phase, several theoretical studies have pointed out that the Dirac points in the spinon band appear due to the existence of a finite gauge field flux in the lattice, as demonstrated in Refs. [*Rev. Mod. Phys.* 78, 17 (2006); *Rep. Prog. Phys.* 80, 016502 (2017); *Rev. Mod. Phys.* 89, 025003 (2017); *Phys. Rev. Lett.* 98, 117205 (2007); *Phys. Rev. X* 7, 031020 (2017); *Phys. Rev. Lett.* 123, 207203 (2019)]. In our case of monolayer 1T-NbSe₂, the quantum spin liquid signatures are observed in the absence of external magnetic field, and hence, the spinon band is not expected to exhibit the Dirac-type dispersion. Moreover, the spinon band is always half filled, so the chemical potential of the Dirac spin liquid should exactly locate at the Dirac point. In such a case, the Dirac spin liquid has the zero spinon density of states which cannot form a spinon Kondo screening cloud. Therefore, our observation of the spinon Kondo resonance peaks at the Hubbard band edges cannot be interpreted by the Dirac spin liquid.

The suggestion of the Z_2 spin liquid phase in bulk 1T-TaS₂ studied in Ref. [*npj Quantum Mater.* 6, 69 (2021)] is interesting, but our monolayer 1T-NbSe₂ differs a lot from the bulk 1T-TaS₂. One essential difference between the Z_2 spin liquid and our proposal of spin liquid with spinon Fermi surface is that the Z_2 spin liquid has the fully gapped spinon spectrum, which means that any spin excitation requires an excitation energy to overcome the spinon gap. As a result, given a Z_2 spin liquid contacted with a metallic state, it cannot give rise to the Kondo resonance at the zero bias. In our experiment, the Kondo resonance peak at zero bias can be clearly observed by placing monolayer 1T-NbSe₂ in contact with a metallic 1H-NbSe₂. This observation definitely rules out the possibility of the gapped Z_2 spin liquid phase. Moreover, a fully gapped Z_2 spin liquid indicates that there are no itinerant spinons

in the ground state, so any spinon Kondo effect is not expected in the fully gapped Z_2 spin liquid, which is quite inconsistent with our experiments.

In the revised manuscript, several sentences have been added on Page 11 to rule out the possibility of the Z_2 QSL and the Dirac QSL. Corresponding references have also been added, according to the Reviewers suggestion.

- (3) The spinon Fermi surface is characterized by a hopping matrix element of order J . What is the order of magnitude for J between these SOD localized spins in 1T-NbSe₂ sample?

Reply: We thank the Reviewer for the question. In our simulation, we take the spinon coupling J between the SOD to be 0.06 eV, which gives the best fit to the electronic DOS measured in our experiments (Fig. 4). Theoretically, the spinon coupling J between the SOD is estimated to have the order of W^2/U , where W represents the band width of the flat band at the Fermi energy and U represents the onsite repulsion interaction. Based on the DFT calculations in Refs. [*Sci. Adv.* 7, eabi6339 (2021)] and [*Nat. Commun.* 12, 1978 (2021)], the band width of the flat band W is in the order of 0.1 eV. The onsite repulsion U is approximated to be the correlated gap size of about 0.4 eV. The estimated J by W^2/U is about 0.03 eV, which is in the same order as we take in the theoretical simulation. We thus conclude that the magnitude for J between these SOD localized spins in monolayer 1T-NbSe₂ is in the range of 0.01 eV \sim 0.1 eV.

In our revised manuscript, we have added the amplitude of the spinon coupling in the Methods, according to the Reviewer's suggestion.

- (4) Authors avoided completely the discussion of the role of disorder. How important is for the emerging QSL in their sample?

Reply: We thank the Reviewer for the question. The effect of disorder on the ground state of a QSL is a fundamental and longstanding issue in the field. Given the

complexity of the QSL, the effect of disorder in the QSL has yet been extensively studied in the literature. In the below, starting from the slave rotor mean field picture, we give a simple analysis on the effect of disorder.

In the slave rotor mean field picture, the random impurities will induce a random distribution of onsite spinon potential. In the weak disorder regime where the energy range of the disorder potential is smaller than the Mott gap, we expect that the induced spinon disorder random potential plays the similar role as that of the weak electric disorders in the metallic phase: the itinerant spinons can get scattered by the spinon disorders, but the spinon Fermi surface remains stable against the weak disorder. In a recent renormalization group study of disorders in QSL [*Phys. Rev. X* 8, 031028 (2018)], it has pointed out that the QSL phase is stable in the regime of weak disorder. In the strong disorder regime where the spatial energy fluctuations of the disorder potential exceed the Mott gap, the disorder potential will strongly affect the Mott insulator ground state, so the slave rotor mean field picture of QSL with spinon Fermi surface is no longer applicable. In our experiment, the size of the correlated gap is in the order of 0.4 eV while no impurity with such energy potential is observed. As a result, we think that our monolayer 1T-NbSe₂ samples are in the weak disorder regime and the QSL ground state remains stable in the weak disorder regime.

In the revised manuscript, several sentences and corresponding references have been added on Page 12 to state that our proposed QSL ground state is stable in the weak disorder regime.

Minor comments:

- (5) I think authors may have overlooked some important experimental papers that previously claimed QSL in 1T-TaS₂. Beside the above mentioned PRX 7, 041054 (2017) and npj Quantum Materials volume 6, 69 (2021), I can also think of Nature Physics 13, 1130-1134 (2017).

Reply: We thank the Reviewer for providing us helpful references. These references

together with the corresponding discussion have been added on Page 10 in our revised manuscript, according to the Reviewer's suggestion.

- (6) Is the fact that they are working on ML and not on a bulk sample relevant? Do they expect similar physics also for the bulk samples?

Reply: We thank the Reviewer for the question. In this work, we focus on monolayer 1T-NbSe₂ samples. The physics for monolayer 1T-NbSe₂ is not suitable for the bulk. The most important reason is that few-layer 1T-NbSe₂ is not a correlated insulator owing to the existence of interlayer coupling. For example, bilayer 1T-NbSe₂ behaves as a metal, as shown in Fig. S4 of our STM and STS measurements.

In our revised manuscript, the sentence "It's worth noting that such a correlated insulating behavior is absent for bilayer 1T-NbSe₂ (actually bilayer 1T-NbSe₂ exhibits a metallic state, see Supplementary Fig. 4), owing to the existence of interlayer coupling" has been added on Page 5. Supplementary Fig. 4 has also been added to demonstrate the electronic properties of bilayer 1T-NbSe₂, according to the Reviewer's question.

Supplementary Fig. 4 | STM image and STS spectra of bilayer 1T-NbSe₂. a, Large-scale STM topographic image of bilayer 1T-NbSe₂ on BLG/SiC(0001) substrate ($V_b = -1.5$ V, $I_t = 10$ pA). b, Typical STS spectra of bilayer 1T-NbSe₂ recorded on the top and hollow sites of CDW pattern, suggesting a metallic state.

(7) Are the quantum fluctuations of the U(1) gauge field the only possible reason for the opening of a gap in the spinon spectrum?

Reply: We thank the Reviewer for the question. In the U(1) spin liquid with spinon Fermi surface, the opening of a gap in the spinon spectrum arises from the instability of the spinon Fermi surface. The instability of the spinon Fermi surface, similar to the case of the electronic Fermi surface, is due to the interactions that involve the itinerant spinons. In general, the quantum fluctuations, or more specifically, the U(1) gauge field fluctuations, are the main mechanisms that give rise to spinon interactions. It is possible that there may exist other mechanisms to gap out the spinon spectrum (e.g., periodic spinon potential that can gap out the spinon spectrum at the Brillouin zone boundary). However, we still think that the quantum fluctuations are the main physical mechanism responsible for the opening of a gap in the spinon spectrum.

Reply to the second Reviewer's report

The authors study monolayer 1H- and 1T-NbSe₂ films using scanning tunneling microscopy and spectroscopy. The experimental data is solid; They observe (1) 3 x 3 CDW on monolayer 1H-NbSe₂ films, (2) the star-of-David pattern with the Mott insulating gap on monolayer 1T-NbSe₂ films, (3) the Kondo peak on 1T/1H-NbSe₂ heterostructures, (4) incommensurate density waves on monolayer 1T-NbSe₂ films, and (5) resonance peaks at the Mott gap edge at magnetic impurities on monolayer 1T-NbSe₂ films.

Before recommending publication, I hope the authors clarify the advances they made in this study.

Among the observations, all (1)-(5) are already known for TaS₂ and TaSe₂ (ref. 14,17,18,25), and (1)-(3) are known for NbSe₂ (ref. 22-24). Given these similarities, (4) and (5) are also highly likely to be observed for NbSe₂. In other words, from these similarities, NbSe₂ is already expected to be a candidate of U(1) QSL if TaS₂ and TaSe₂ are so. Under this circumstance, the authors actually observe (4) and (5) and interpret them in the same manner as TaS₂ and TaSe₂. I understand that (4) and (5) have never been observed in NbSe₂. Meanwhile, what do we learn from the confirmation of expected behaviors?

Reply: We thank the Reviewer for the careful consideration of our manuscript and the positive comment “*The experimental data is solid*”. QSL has been intensely investigated since the first prediction in 1973. During the last decades, many theoretical predictions have been focused on the QSL models of 2D triangular-lattice correlated insulators. Experimentally, several single-crystal and organic materials with triangular atomic/molecular layers have been demonstrated to be QSL candidates, such as YbMgGaO₄ [*Nature* 540, 559 (2016)], EtMe₃Sb[Pd(dmit)₂]₂ [*Science* 328, 1246 (2010)], and 1T-TaS₂ [*Nat. Phys.* 13, 1130 (2017); *npj Quantum Mater.* 6, 69 (2021)].

Until 2021, monolayer 1T-TaSe₂ was verified to be the first QSL candidate in the 2D limit, and has been the only one up till now. Searching for new QSL candidates in real 2D systems and revealing the nature of QSL physics are of great significance.

In this work, we provide the first experimental evidence of the QSL signatures in monolayer 1T-NbSe₂, another 2D system exhibiting an isostructural compound with monolayer 1T-TaSe₂. Indeed, most of our observations in monolayer 1T-NbSe₂ are similar as those reported in monolayer 1T-TaSe₂. Even so, **we would like to point out that the QSL behavior in monolayer 1T-NbSe₂ should not be directly viewed as a simple extrapolation from 1T-TaSe₂ to 1T-NbSe₂ by intuition, but needs to be carefully investigated, because there are still many differences in electronic properties among 1T-NbSe₂, 1T-TaS₂, and 1T-TaSe₂, as summarized in Supplementary Tab. 1.** For example, *for the bilayer system*, bilayer 1T-NbSe₂ is usually a metal, while bilayer 1T-TaSe₂ is still a Mott insulator with a reduced energy gap. No experimental results have been reported on the electronic structures of bilayer 1T-TaS₂. *For the surface states of the bulk*, 1T-TaSe₂ and 1T-TaS₂ can be an insulator or a metal, dependent on the interlayer stacking. However, bulk 1T-NbSe₂ has not been successfully synthesized up till now. Since the bilayer and bulk samples of 1T-NbSe₂, 1T-TaS₂, and 1T-TaSe₂ exhibit completely different properties, the QSL state in monolayer 1T-NbSe₂ cannot be deduced only by the correlated insulating behaviors before the report of our experimental results.

Moreover, our work not only reveals the spatially resolved spinon Kondo effect modulated by the gauge binding interaction, but also extends the spinon Kondo effect into a high spin regime. Comparing with Ref. [*Nat. Phys.* 18, 1335 (2022)] by using Co atoms with $S = 1/2$ spin as local probes, in our experiments, the MnPc molecules with $S = 3/2$ spin are used to investigate the QSL signatures in monolayer 1T-NbSe₂. Different from the Kondo screening regime with $S = 1/2$ spin, in such an under-screened Kondo regime with $S = 3/2$ spin, a local spin of $S = 1$ is expected to remain in the magnetic impurity, while the spin $S = 1/2$ undergoes Kondo spin exchange with the itinerant spinons.

One important point we can learn from the confirmation of the correlated insulating behavior and the QSL state in monolayer 1T-NbSe₂ is that the correlated insulating gap is not empty. Instead, both the super-modulation and the resonance peaks at the Hubbard band edges indicate that some exotic states locate inside the charge gap of monolayer 1T-NbSe₂. Therefore, our results indicate the significance of further probing the in-gap exotic states in monolayer 1T-NbSe₂.

Another important point obtained from our work is to carry out a further study of the heterostructures formed by 1T-NbSe₂ and 1H-NbSe₂ as well as their isostructural compounds. Below $T_c \sim 2.4$ K, monolayer 1H-NbSe₂ is known to be an Ising superconductor that is resilient to a strong in-plane upper critical field [*Nat. Phys.* 12, 139 (2016); *Nano Lett.* 17, 6802 (2017)]. Based on the confirmed QSL state in a correlated insulator monolayer 1T-NbSe₂, it will be extremely interesting to study the interplay between the Ising superconductivity and the QSL-type correlated insulating state. Recently, superconducting boundary modes [*Nat. Phys.* 17, 1413 (2021)] and magnetic memory phenomenon [*Nature* 607, 692 (2022)] have been reported in 1T-TaS₂/1H-TaS₂ heterostructures, however, the true ground state in such heterostructures remains under debate. Our confirmation of the QSL-type correlated insulating states in monolayer 1T-NbSe₂ thus provides a new platform to explore unexpected novel quantum phenomena arising from the interplay between strong electronic correlation and superconductivity.

In a brief summary, our experimental observation of the QSL signature in a correlated insulator monolayer 1T-NbSe₂ demonstrates that monolayer 1T-NbSe₂ is the second QSL candidate in real 2D systems and can be regarded as a new platform to study the correlated insulating states in 2D van der Waals quantum materials. Moreover, the QSL-based novel quantum phenomena such as the spinon Kondo effect with a high spin regime have been experimentally verified. We believe that the QSL signature exhibited in monolayer 1T-NbSe₂ lay the foundation in the future study of the strong electronic correlations.

We appreciate the constructive comment from the Reviewers, which are of great

help for improving the quality of our work. According to the Reviewer’s valuable comment, the sentences “Since the QSL states were first predicted in 1973, many theoretical predictions have been focused on the QSL models of 2D triangular-lattice correlated insulators. Experimentally, several single-crystal and organic materials with triangular atomic or molecular layers have been demonstrated to be QSL candidates^{10,11,43,44}. Until 2021, monolayer 1T-TaSe₂ was verified to be the first QSL candidate in the 2D limit¹⁹. In this work, we provide the experimental evidence of the QSL signatures in monolayer 1T-NbSe₂, another 2D system exhibiting an isostructural compound with monolayer 1T-TaSe₂. Indeed, most of our observations in monolayer 1T-NbSe₂ are similar as those reported in monolayer 1T-TaSe₂. Even so, the QSL behavior in monolayer 1T-NbSe₂ should not be directly viewed as a simple extrapolation from 1T-TaSe₂ to 1T-NbSe₂ by intuition, but needs to be carefully investigated, because there are still many differences in electronic properties^{10,19,20,27,45,46}, as summarized in Supplementary Tab. 1” and the corresponding references have been added on Pages 10 and 11 to emphasize the importance and significance of our work. The sentences “...in 2D quantum materials, particularly including the in-gap exotic states and the QSL-superconductivity interaction in van der Waals heterostructures⁵⁵⁻⁵⁸. Moreover, the collective nature of QSL in triangular lattice with SFS ground state can be further probed, as theoretically predicted very recently^{59,60}” have been added on Pages 12 and 13 to demonstrate the further research directions. Supplementary Tab. 1 has also been added to summarize the discrepancies among 1T-NbSe₂, 1T-TaS₂, and 1T-TaSe₂. We hope our response can satisfactorily address the comment.

Supplementary Tab. 1 | Summarization of discrepancies among 1T-NbSe₂, 1T-TaS₂, and 1T-TaSe₂.

	1T-NbSe ₂	1T-TaS ₂	1T-TaSe ₂
Bi-layer	Metal  [this work]	Not reported	Mott insulator  [Nat. Phys. 17, 1154 (2021)]
Surface states for bulk	Not reported	Metal/insulator (Stacking dependent)  [Nat. Commun. 11, 2477 (2020)]	Metal/insulator (Stacking dependent)  [Phys. Rev. B 106, 075153 (2022)]
QSL in mono-layer	U(1) QSL [this work]	Not reported	U(1) QSL [Nat. Phys. 17, 1154 (2021)]
QSL in bulk	Not reported	Z₂ QSL [npj Quantum Mater. 6, 69 (2021)]	Not reported
Spinon Kondo effect	Monolayer, molecule with $S = 3/2$ [this work]	Not reported	Monolayer, atom with $S = 1/2$ [Nat. Phys. 18, 1335 (2022)]

Reply to the third Reviewer's report

The authors prepared a monolayer 1T-NbSe₂ film by MBE and investigated its electronic structures by STM/STS. 1T-NbSe₂ is naturally expected to possess similar electronic structures to its sister compounds 1T-TaSe₂ and 1T-TaS₂, well-known candidates for the quantum spin liquid. The authors observed a Kondo-like resonance in monolayer 1T-NbSe₂ placed on metallic 1H-NbSe₂, which suggests the local spin at the Star-of-David CDW cluster. They also observed the incommensurate spatial modulations in the electronic state of 1T-NbSe₂ on graphene, which may be related to the spinon Fermi surface in the quantum spin liquid. In addition, the authors observed a pair of resonance peaks in the tunneling spectra taken near the MnPc molecule deposited on 1T-NbSe₂. They state that the observed peaks can be explained by the model based on the quantum spin liquid. Overall, the authors claim that 1T-NbSe₂ is a new promising quantum spin liquid material.

The Kondo-like resonance and the incommensurate electronic state modulations have already been observed in 1T-TaSe₂, and the discussion given in this manuscript basically follows that of the previous publications. The effect of the deposited MnPc molecule is new and interesting. However, I think that the reported results are still premature. I recommend that the authors shorten the discussion on the Kondo resonance and the incommensurate modulation and perform more systematic experiments and theoretical analyses on the effects of magnetic molecules. More specific comments are listed below.

Reply: We thank the Reviewer for the positive comment “*The effect of the deposited MnPc molecule is new and interesting*”. We also appreciate the Reviewer for the valuable comments especially the effects of magnetic molecules on a QSL state. We have carefully revised our manuscript according to the Reviewer’s valuable suggestions and comments, which are of great help for improving the quality of our work. We hope

our response below can satisfactorily address all the comments.

(1) I recommend that the authors explain why the UHB splits into UHB₁ and UHB₂.

Reply: We thank the Reviewer for the valuable suggestion. The splitting of the UHB into UHB₁ and UHB₂ in monolayer 1T-NbSe₂ is attributed to the spatially varying Coulomb repulsion and the reduced screening in two dimensions, which is in accordance with monolayer 1T-TaSe₂ that was previously demonstrated in Ref. [*Nat. Phys.* 16, 218 (2020)] (Fig. R2a).

Specifically, during the process of electrons injecting from a STM tip into monolayer 1T-NbSe₂, the LHB preferentially occupies the orbital at the center of SOD, thus creating a spatially varying Coulomb repulsion landscape that affects the distribution of the UHB spectral weight (schematically shown in Fig. R2b, extracted from [*Nat. Phys.* 16, 218 (2020)]). Considering the composite nature of the UHB (contributed by 13 Nb atoms in each SOD), the UHB₁, UHB₂, and their orbital textures at can be regarded as a redistribution of the UHB spectral density (Fig. R2c), in response to the enhanced Coulomb interactions that arise from reduced screening in two dimensions.

In our revised manuscript, Fig. R2c has been added as Supplementary Fig. 5 to exhibit the orbital textures of LHB & VB, UHB₁ and UHB₂ in monolayer 1T-NbSe₂. The sentence “The splitting of the UHB into UHB₁ and UHB₂ is attributed to the spatially varying Coulomb repulsion and the reduced screening in 2D systems, which is consistent with monolayer 1T-TaSe₂ that was previously demonstrated^{19,20}” has been added on Page 5 to explain the reason why the UHB splits into UHB₁ and UHB₂, according to the Reviewer’s suggestion.

Fig. R2. Coulomb interaction and orbital textures. a, Typical STS spectrum of monolayer 1T-TaSe₂, extracted from Ref. [Nat. Phys. 16, 218 (2020)]. b, Schematic of LHB-induced Coulomb repulsion, extracted from Ref. [Nat. Phys. 16, 218 (2020)]. c, STM image and orbital textures of LHB & VB, UHB₁ and UHB₂ in monolayer 1T-NbSe₂ in our experiments.

(2) In Fig. 1, each Star-of-David CDW cluster is imaged as a triangular object, which apparently breaks inversion symmetry. Please explain the reason for this.

Reply: We thank the Reviewer for the question. Supplementary Fig. 2a,b shows top and side views of atomic and SOD structures of monolayer 1T-NbSe₂, where top/bottom Se atoms are exhibited in pink/orange. From Supplementary Fig. 2c we can find out the topmost Se layer exhibits a triangular object in each SOD motif. Considering that STM images mainly reflect the information about the topmost Se layer, in such a case, each SOD motif is expected to be imaged as a triangular object (Supplementary Fig. 2d).

In our revised manuscript, the sentences “...in the middle layer, accompanied by six top-layer Se atoms and six bottom-layer Se atoms. From the atomic-resolution STM image shown in the inset of Fig. 1b and Supplementary Fig. 2, we can find out that each SOD motif exhibits as a triangle, because the six top-layer Se

atoms have more contribution to the STM images” have been added on Page 4. Supplementary Fig. 2 has also been added, according to the Reviewer’s suggestion.

- (3) The authors mention that they can rule out the influences of the graphene substrate and the tip effect as the origin of the observed incommensurate electronic-state modulations. I recommend that the authors show the corresponding experimental data in the supplementary information.

Reply: We thank the Reviewer for the valuable comment and suggestion.

Firstly, we rule out the possibility of the graphene substrate as the origin of the incommensurate electronic modulations. According to the Ref. [*Nat. Phys.* 8, 382 (2012)], the wavelength of the moiré pattern constructed by 1T-NbSe₂ and graphene can be obtained by

$$\lambda = \frac{(1 + \delta)a_G}{\sqrt{2(1 + \delta)(1 - \cos \phi) + \delta^2}}$$

where $a_G \approx 0.246$ nm is the lattice constant of graphene, $a_{NbSe_2} \approx 0.346$ nm is the lattice constant of 1T-NbSe₂ [*Phys. Rev. Lett.* 121, 026401 (2018)], and the lattice mismatch between 1T-NbSe₂ and graphene $\delta = (a_{NbSe_2} - a_G)/a_G \approx 0.407$. The relative rotation angle ϕ between 1T-NbSe₂ and graphene lattices shown in Fig. 1b is about 28°, according to the atomic-resolution STM images shown in Supplementary Fig. 6. In such a case, the wavelength of the moiré pattern $\lambda \approx 0.491$ nm, which is quite inconsistent with the observed incommensurate electronic-state modulations. Moreover, we develop an in-situ STM manipulation technique to precisely control the relative angle between 1T-NbSe₂ and graphene lattices, as shown in Supplementary Fig. 7. We find out that the observed incommensurate electronic modulations are invariable under different relative angles, and are not derived from the graphene substrate (Supplementary Fig. 8). Therefore, we can rule out the possibility of the graphene substrate as the origin of the incommensurate electronic-state modulations.

Secondly, we rule out the possibility of the tip effect as the origin of the incommensurate electronic modulations. On the one hand, during our measurements, the incommensurate electronic modulations only appear when the recorded energies are close to the Hubbard band edges of monolayer 1T-NbSe₂. On the other hand, we carry out the STS maps of monolayer 1H-NbSe₂ (a CDW metal) on graphene under the same tip conditions. We find out that the STS map recorded at the energy of -0.2 eV exhibits no additional electronic modulation, as shown in Supplementary Fig. 9. These phenomena highlight that the incommensurate electronic modulations are the intrinsic electronic properties of monolayer 1T-

NbSe₂ near the Hubbard band edges, helping us rule out the possibility of the tip effect as the origin of the incommensurate electronic modulations.

In our revised manuscript, Supplementary Figs. 6-9 and several sentences, as well as the corresponding references, have been added in the Supplementary Information to rule out the possibility of the graphene substrate and the tip effect as the origin of the incommensurate electronic modulations, according to the Reviewer's valuable comment and suggestion.

Supplementary Fig. 6 | Atomic-resolution STM image of monolayer 1T-NbSe₂ on graphene. a, Large-scale STM topographic image of monolayer 1T-NbSe₂ on graphene. b, Atomic-resolution STM image of 1T-NbSe₂. In each SOD motif, the top Se atoms dominate the STM images as a triangular bright protrusion. The basis vectors of 1×1 atomic lattices and $(\sqrt{13} \times \sqrt{13}) R 13.9^\circ$ CDW lattices are marked by red and black arrows, respectively. c, Atomic-resolution STM image of graphene. The basis vectors of 1×1 atomic lattices are marked by white arrows.

Supplementary Fig. 7 | Movement of monolayer 1T-NbSe₂ on BLG via an in-situ STM manipulation technique. The atomic orientations between monolayer 1T-NbSe₂ and BLG can be arbitrary during our manipulation, which can be captured by STM images. Here the red dotted outline donates the initial location of the central monolayer 1T-NbSe₂, and the black dotted outline donates its location after the manipulation.

Supplementary Fig. 8 | Incommensurate electronic modulations of monolayer 1T-NbSe₂. a, STM image of another monolayer 1T-NbSe₂ recorded at the sample bias of -1 V. b, FFT image of the STS map at the sample bias of -1.0 V. The white hexagon represents the CDW Brillouin zone. The six bright spots enclosed by the red circles indicate the CDW wavevectors of monolayer 1T-NbSe₂. c, STS map of monolayer 1T-NbSe₂ recorded at the sample bias of 0.2 V with the same location as panel a. d, FFT image of panel c. The spots enclosed by the yellow circles are related to the charge modulation with the wavelength larger than $\sqrt{3}$ times of the CDW wavevectors. e, STS map of graphene substrate recorded at the sample bias of 0.2 V. f, FFT image of panel e. The outer (inner) six bright spots enclosed by the green (orange) circles indicate the graphene lattice (graphene-SiC moiré superlattice).

Supplementary Fig. 9 | Electronic properties of monolayer 1H-NbSe₂. a, STM image of monolayer 1H-NbSe₂. b, Corresponding STS map recorded at the sample bias of -0.2 V. c, Corresponding FFT images of the STS map. The outer (inner) six bright spots enclosed by the white (red) circles indicate the 1×1 atomic (3×3 CDW) wavevectors of monolayer 1H-NbSe₂. There is no additional long-range charge modulation with the wavelength larger than the CDW wavevectors, indicating the absence of spinon excitations.

- (4) In Fig. 4a and 4b, the MnPc-induced peaks are subtle, and it is unclear for me to recognize the site dependence argued in the text. I recommend that the authors quantify the peak intensities and show their spatial variations in 2D images to demonstrate the site dependence clearly.

Reply: We thank the Reviewer for the valuable comment. Here we take the resonance peak appearing near the UHB edge as an example to demonstrate the peak intensities and the spatial variations, because the peak near the LHB edge is usually embedded into the LHB&VB peak, which may influence the captured peak intensities. Supplementary Fig. 12 shows a typical STM image of the MnPc-top configuration and the corresponding STS map at the resonance peak energy near the UHB edge, where the center Mn ion locates on a SOD motif, most of the right-Pc locates on a SOD motif, and the up-, bottom-, and left-Pc locate off SOD motifs. From Supplementary Fig. 12, we can find out that the resonance peak near the UHB edge of the MnPc-top configuration mainly locates at the center Mn ion and the right-Pc positions, with the intensity ratio between the center Mn and the up-Pc of about 2 (It's worth noting that STS mappings can only reflect the relative intensities at specific energies). Such a result implies that the overlap between the holon state in the QSL monolayer 1T-NbSe₂ and the magnetic moment of the Mn/right-Pc position is larger than that of the other three Pc positions.

Supplementary Fig. 13 shows a typical STM image of the MnPc-hollow configuration and the corresponding STS map at the resonance peak energy near the UHB edge, where the both the center Mn ion and the four Pc positions locate off SOD motifs. From Supplementary Fig. 13, we can find out that the resonance peak near the UHB edge of the MnPc-top configuration mainly locates at the center Mn ion position, with the peak intensity ratio between the center Mn and the up-left-Pc of about 4, implying the dominate overlap between the holon state in the QSL monolayer 1T-NbSe₂ and the magnetic moment of the Mn ion.

In our revised manuscript, Supplementary Figs. 12 and 13 as well as several sentences have been added to demonstrate the peak intensities and the spatial variations of the resonance peak appearing near the UHB edge, according to the Reviewer's suggestion.

Supplementary Fig. 12 | Spatially resolved intensities of the resonance peak appearing near the UHB edge for the MnPc-top configuration. a,b, Topographic STM image of a MnPc molecule adsorbed on a SOD motif. c,d, Corresponding STS map at the resonance peak energy of 0.65 eV, reflecting the spatial variations of the spinon Kondo effect.

Supplementary Fig. 13 | Spatially resolved intensities of the resonance peak appearing near the UHB edge for the MnPc-hollow configuration. a,b, Topographic STM image of a MnPc molecule adsorbed off a SOD motif. c,d, Corresponding STS map at the resonance peak energy of 0.65 eV, reflecting the spatial variations of the spinon Kondo effect.

- (5) The band-edge resonances due to the spinon Kondo effect occur near the top of the holon band and the bottom of the doublon band. If I understand correctly, the holon and the doublon bands are essentially LHB and UHB, respectively. In the experiment, one peak indeed appears near the top of the LHB, but another peak is observed near 0.7 eV, which is higher than both UHB_1 and UHB_2 in the pristine regime. Does the MnPc modify the Mott gap to produce a single UHB at higher energy? If so, please explain the mechanism in detail.

Reply: We thank the Reviewer for the valuable comment. The Reviewer is right that in our experiment the resonance peak from the UHB appears near 0.7 eV, which

is a bit higher than the UHB₁ and UHB₂ energy. For the reason behind, we think it is probably due to the MnPc molecule's modification on the UHB states, which also agrees with the Reviewer's conjecture.

In principle, for the idealized magnetic impurity model we constructed in the main text, the spinon Kondo effect induced a pair of resonance peaks, which should occur at the top of the LHB and the bottom of the UHB, respectively. However, in the reality case of magnetic MnPc molecule, the MnPc molecule has the spatial spread, so the charge distribution on the MnPc molecule will modify the original charge distribution from the LHB electronic states. In such a case, the UHB states injected from the STM tip must feel a different profile of Coulomb repulsions from that of pristine monolayer 1T-NbSe₂. As a result, the UHB states in the presence of a MnPc molecule get the spectral weight re-distributed. Such redistribution of the UHB spectral weight makes the spinon Kondo effect induced upper resonance peak move to a higher energy than the original UHB₂ and UHB₁ energy.

The complicated spatially varying orbitals of the MnPc molecule makes it difficult for us to simulate the multi-orbital Anderson impurity model on a quantum spin liquid, so the very detail mechanism about the upper resonance peak with its energy higher than UHB₁ and UHB₂ is not known. However, we still strongly believe that the interplay between the MnPc and the charge distribution of the LHB electronic states gives the reason why the upper resonance peak appears at a higher energy than the UHB₁ and UHB₂ in the pristine sample. In Fig. 4a,b, it can be seen that the MnPc molecule has little effect on the LHB energy, and the main effect of the MnPc molecule is seen to change the UHB₁ and UHB₂ states, especially for the spectrum measured at the hollow position of the 1T-NbSe₂. The change of the UHB₁ and UHB₂ states can only be attributed to the MnPc molecule modified LHB states induced Coulomb repulsion.

In the revised manuscript, several sentences have been added on Pages 9 and 10 to explain that the MnPc molecule is supposed to modify the UHB state and gives rise to the upper resonance peak near 0.7 eV.

(6) I recommend that the authors experimentally and theoretically investigate how the band-edge resonances change when other metal phthalocyanine molecules are deposited. This experiment should be technically simple and should provide information about the spinon Kondo effect for different spins.

Reply: We thank the Reviewer for the valuable comment and suggestion. The spinon Kondo effect arises from the spin exchange between the itinerant spinons and the local spin-1/2 at the impurity. In the case of spin-3/2, the local spin-1/2 gets screened by Kondo cloud formed by the spinons, and the remaining local spin with $S = 1$ is decoupled. In the case of spin-1/2, the local spin gets fully screened by the spinon Kondo cloud. In the case of spin-0, there is no spin exchange between the magnetic impurity and the itinerant spinons. As a result, one can see that the spinon Kondo effect for different spins differs in the Kondo screening. For the spinon Kondo resonance peaks observed in our experiment, the resonance peaks are due to the joint effect of the spinon Kondo coupling and the gauge binding effect. As long as the local spin at the impurity has finite coupling with the itinerant spinons, sufficiently large gauge binding interaction can always give rise to the band edge resonance peaks. As a result, we expect that the band edge resonance peaks appear at the magnetic impurity, but disappear if there is no local spin at the impurity.

As the Reviewer's suggestion, we also carry out STM and STS measurements of a non-magnetic molecule ZnPc on monolayer 1T-NbSe₂. As summarized in Supplementary Fig. 14, no additional resonance peaks appear at the band edges regardless of atomic configurations (ZnPc-top or ZnPc-hollow), implying the absence of the spinon Kondo effect for spin-0 regime.

In our revised manuscript, several sentences have been added on Page 9 to demonstrate the experimental results of a non-magnetic molecule ZnPc on monolayer 1T-NbSe₂. Supplementary Fig. 14 and several sentences have been added on Pages S15, S17, and S18 of Supplementary Information to analyze the spinon Kondo effect for different spins.

Supplementary Fig. 14 | Atomic and electronic structures of a ZnPc molecule on monolayer 1T-NbSe₂. a-d, STM images and atomic structures of a ZnPc molecule on monolayer 1T-NbSe₂, with the central Zn atom on the top and hollow sites of a SOD motif, respectively. e,f, STS spectra recorded on pristine monolayer 1T-NbSe₂ as well as on Zn atom of the ZnPc-top and ZnPc-hollow. No additional resonance peaks appear at the band edges, implying the absence of the spinon Kondo effect.

List of changes:

1. On Page 4, the sentences "...in the middle layer, accompanied by six top-layer Se atoms and six bottom-layer Se atoms. From the atomic-resolution STM image shown in the inset of Fig. 1b and Supplementary Fig. 2, we can find out that each SOD motif exhibits as a triangle, because the six top-layer Se atoms have more contribution to the STM images" have been added to demonstrate the reason why each SOD motif is imaged as a triangular object.
2. Supplementary Fig. 2 has been added to demonstrate the reason why each SOD motif is imaged as a triangular object.
3. On Page 5, the sentences "The splitting of the UHB into UHB₁ and UHB₂ is attributed to the spatially varying Coulomb repulsion and the reduced screening in 2D systems, which is consistent with monolayer 1T-TaSe₂ that was previously demonstrated^{19,20}. In addition, the UHB₂ is highly hybridized with the UHB₁ because of the composite nature of the UHB (contributed by 13 Nb atoms in each SOD)" have been added to demonstrate the reason of the UHB splitting.
4. On Page 5, the sentences "It's worth noting that such a correlated insulating behavior is absent for bilayer 1T-NbSe₂ (actually bilayer 1T-NbSe₂ exhibits a metallic state, see Supplementary Fig. 4), owing to the existence of interlayer coupling" have been added to demonstrate the electronic properties of bilayer 1T-NbSe₂.
5. Supplementary Fig. 4 has been added to demonstrate the electronic properties of bilayer 1T-NbSe₂.
6. On Page 5, the sentences "Moreover, the intensities of these DOS peaks exhibit remarkably spatial dependency. As we can see from the spatially resolved STS spectra...in accordance with previously reported^{23-26,29}. Such a result can also be intuitively presented from the orbital textures. Here we show the measured STS maps at the Hubbard band energies of monolayer 1T-NbSe₂ in Supplementary Fig. 5, and compare the orbital textures of monolayer 1T-NbSe₂ to those of monolayer 1T-TaSe₂²⁰ and bulk 1T-TaS₂³¹. For the LHB energy, the STS maps of all the three

systems display the same patterns where the electrons concentrating at the center of SOD motifs. In contrast, for the UHB energy, the STS maps of bulk 1T-TaS₂ exhibit similar patterns to those of LHB, while they exhibit completely different patterns of monolayer 1T-NbSe₂ and monolayer 1T-TaSe₂ where electrons prefer to locate around the outer rim of SOD motifs” have been added to describe the orbital textures of monolayer 1T-NbSe₂ and comparing them to monolayer 1T-TaSe₂ and bulk 1T-TaS₂.

7. Supplementary Fig. 5 has been added to demonstrate the orbital textures in monolayer 1T-NbSe₂.
8. On Pages 5 and 6, the sentences “Previously, it has been theoretically demonstrated that the interaction between the localized and itinerant orbitals plays a dominant role in the electronic structures of bulk 1T-TaS₂³¹. Our results obviously verify that, on the basis of such a two-orbital model and density functional theory (DFT) + *U* simulations^{32,33}, additional Coulomb interactions should also be taken into account to explain the spatial repulsion of LHB and UHB in monolayer 1T-NbSe₂ and monolayer 1T-TaSe₂³¹. Although bulk 1T-TaS₂ is usually considered to be a quasi-2D system, a slight interlayer interaction may increase electronic delocalization and screening, thus resulting in a reduction of Coulomb interactions^{20,31}” have been added to explain the spatial distribution of the UHB.
9. Supplementary Figs. 6-9 have been added to rule out the possibility of the graphene substrate and the tip effect as the origin of the incommensurate electronic modulations.
10. Supplementary Figs. 12 and 13 have been added to demonstrate the peak intensities and the spatial variations of the resonance peak appearing near the UHB edge.
11. On Pages 9 and 10, the sentences “In reality, since magnetic MnPc molecules have a finite spatial spread to modify the Coulomb repulsion profile and the spin moment distribution in monolayer 1T-NbSe₂, the magnetic coupling between the MnPc magnetic moments and the spins in monolayer 1T-NbSe₂ SOD motifs...It

is possible that the coupling between the MnPc and the SOD in 1T-NbSe₂ changes the charge distribution of the LHB, so the Coulomb repulsion acted on the injected electronic states from the UHB becomes different from that in the pristine case²⁰. As a result, the original UHB₁ and UHB₂ may undergo a spectral weight re-distributed. We can see from Fig. 4” have been added to demonstrate the reason of MnPc-modulated UHB state.

12. Supplementary Fig. 14 has been added to demonstrate the experimental results of a non-magnetic molecule ZnPc on monolayer 1T-NbSe₂.
13. On Pages 10 and 11, the sentences “Since the QSL states were first predicted in 1973, many theoretical predictions have been focused on the QSL models of 2D triangular-lattice correlated insulators. Experimentally, several single-crystal and organic materials with triangular atomic or molecular layers have been demonstrated to be QSL candidates^{10,11,43,44}. Until 2021, monolayer 1T-TaSe₂ was verified to be the first QSL candidate in the 2D limit¹⁹. In this work, we provide the experimental evidence of the QSL signatures in monolayer 1T-NbSe₂, another 2D system exhibiting an isostructural compound with monolayer 1T-TaSe₂. Indeed, most of our observations in monolayer 1T-NbSe₂ are similar as those reported in monolayer 1T-TaSe₂. Even so, the QSL behavior in monolayer 1T-NbSe₂ should not be directly viewed as a simple extrapolation from 1T-TaSe₂ to 1T-NbSe₂ by intuition, but needs to be carefully investigated, because there are still many differences in electronic properties^{10,19,20,27,45,16}, as summarized in Supplementary Tab. 1” have been added to demonstrate the significance of our work.
14. On Page 11, the sentences “Our experimental phenomena can further help us rule out the possibility of other QSL states in monolayer 1T-NbSe₂, such as the Z₂ QSL state and the Dirac QSL. Although Z₂ QSL state was confirmed in the isostructural bulk 1T-TaS₂^{10,11}, it is unlikely to be the ground state of monolayer 1T-NbSe₂. One essential reason is that the Z₂ QSL has a fully gapped spinon spectrum, any spin excitation requiring an excitation energy to overcome the spinon gap. In such a case, the Kondo resonance should not occur at the Fermi energy as a Z₂ QSL

contacting with a metal. Moreover, a fully gapped Z_2 QSL indicates that there are no itinerant spinons in the ground state, so any spinon Kondo effect is not expected. In our experiment, the Kondo resonance peak at the Fermi energy can be clearly observed by placing monolayer 1T-NbSe₂ onto a metallic 1H-NbSe₂, and the spinon Kondo effect can be observed by depositing a magnetic impurity onto monolayer 1T-NbSe₂. Based on our experimental observations, we can definitely rule out the possibility of the gapped Z_2 QSL state” have been added to rule out the possibility of the Z_2 QSL in monolayer 1T-NbSe₂.

15. On Page 11, the sentences “Besides the U(1) QSL, the Dirac QSL is also allowed to exhibit gapless spin excitations⁴⁷⁻⁵¹. However, previous predictions proposed that an external gauge field flux is usually required to stabilize the Dirac QSL⁴⁷⁻⁵¹. Since our measurements are all carried out in the absence of magnetic field, the gapless spin excitations indicated by our observations in monolayer 1T-NbSe₂ are more likely from the SFS rather than the Dirac-type spinon spectrum. Therefore, our results strongly suggest that monolayer 1T-NbSe₂ is a U(1) QSL with SFS” have been added to rule out the possibility of the Dirac QSL in monolayer 1T-NbSe₂.
16. On Page 12, the sentences “Our proposed ground state of QSL with SFS is also stable against weak disorders⁵². In the slave rotor mean field picture⁵³, the random distributed impurities will induce disorder potentials in the spinon channel. In the weak disorder regime where the impurity potential range is smaller than the correlated gap size⁵², the spinon disorders are expected to play the similar role as that of the ordinary disorders in a metal⁵⁴. In our monolayer 1T-NbSe₂ samples, there is only a low concentration of atomic defects, the energy potential of which cannot be comparable to the correlated gap size, indicating that our samples are in the weak disorder regime. Since the ground state of SFS is protected by the correlated gap, the weak disorder potentials may not affect the nature of the emerging QSL proposed in our samples” have been added to demonstrate the QSL with SFS is stable against weak disorders.
17. On Pages 12 and 13, the sentences “...in 2D quantum materials, particularly

- including the in-gap exotic states and the QSL-superconductivity interaction in van der Waals heterostructures⁵²⁻⁵⁵. Moreover, the collective nature of QSL in triangular lattice with SFS ground state can be further probed, as theoretically predicted very recently^{56,57} have been added to demonstrate the further investigations on the QSL physics.
18. On Page 18, the sentence “spinon coupling between the nearest-neighbor SOD motifs” has been added to demonstrate the hopping parameters. Corresponding description of theoretical calculations have also been added in the Supplementary Information.
 19. Refs. [*npj Quantum Mater.* 6, 69 (2021)], [*Nat. Phys.* 13, 1130 (2017)], [*Phys. Rev. X* 7, 041054 (2017)], and [*Phys. Rev. B* 106, 075153 (2022)] have been added to demonstrate the QSL signatures and orbital textures in bulk 1T-TaS₂ /1T-TaSe₂.
 20. Refs. [*Phys. Rev. B* 90, 045134 (2014)] and [*Phys. Rev. B* 105, L081106 (2022)] have been added to introduce the DFT + U theoretical model.
 21. Ref. [*Nat. Phys.* 8, 382 (2012)] has been added to rule out the possibility of the graphene substrate and the tip effect as the origin of the incommensurate electronic modulations.
 22. Refs. [*Science* 328, 1246 (2010)], [*Nature* 540, 559 (2016)], and [*Nat. Commun.* 11, 2477 (2020)] have been added to introduce the existing QSL candidates.
 23. Refs. [*Rev. Mod. Phys.* 78, 17 (2006)], [*Phys. Rev. Lett.* 98, 117205 (2007)], [*Rev. Mod. Phys.* 89, 025003 (2017)], [*Rep. Prog. Phys.* 80, 016502 (2017)], and [*Phys. Rev. Lett.* 123, 207203 (2019)] have been added to introduce the U(1), Z₂, and Dirac QSL physics.
 24. Refs. [*Phys. Rev. X* 8, 031028 (2018)], [*Phys. Rev. B* 70, 035114 (2004)], and [*Phys. Rev. Research* 2, 013099 (2020)] have been added to demonstrate QSL with SFS under weak disorders.
 25. Refs. [*Nat. Phys.* 12, 139 (2016)], [*Nano Lett.* 17, 6802 (2017)], [*Nat. Phys.* 17, 1413 (2021)], and [*Nature* 607, 692 (2022)] have been added to demonstrate the further investigations on the QSL physics.

REVIEWER COMMENTS

Reviewer #1 (Remarks to the Author):

Authors have extensively and convincingly replied to my comments. Especially, the comparison between 1T-TaS₂, 1T-TaSe₂ and 1T-NbSe₂ (this work) in light of different orbital textures is interesting and relevant. Next, I find the discussion of the possible QSL states also interesting. By performing experiments on different substrates indeed helps to elucidate the best QSL candidate state.

In conclusion, the present submission and the extensive reply fully meet the high standards of Nature Communication journal. Therefore, I recommend this submission for a publication.

Reviewer #2 (Remarks to the Author):

The author emphasizes the differences from the bulk, but it seems to me that this is a repetition of what has already been reported. On the other hand, given that certain discussion have been added, it would be worthwhile to publish it and solicit a wide range of opinions.

Reviewer #3 (Remarks to the Author):

The authors have addressed some of the issues that I raised in the previous report. However, the authors' replies on the following three issues are not convincing enough.

1. The results on Kondo-like resonance and the incommensurate electronic state modulations are still overemphasized. Even if there are potential differences between 1T-NbSe₂ and 1T-TaSe₂, the presented results do not give any insight into the differences. Again, I recommend that the authors shorten (or even omit) the discussion on the Kondo resonance and the incommensurate modulation and perform more systematic experiments and theoretical analyses on the effects of magnetic molecules.

2. The authors suggested that the MnPc molecule could change the potential profile. Although I understand that it is difficult to quantitatively explain the effect of a modified potential profile, the authors should qualitatively explain why only one UHB appears instead of UHB1 and UHB2 and why the apparent Mott gap increases on the MnPc molecule.

3. The authors claim that non-magnetic ZnPc molecules do not produce spinon Kondo peaks. In Supplementary Fig. 14 f, I see a shoulder just below the UHB that is similar to the putative spinon Kondo peak on the MnPc.

Reply to Reviewers' comments and suggestions

We thank all the three Reviewers for their comments and suggestions for a better presentation of the paper. We have carefully considered all the comments. In the following, we reply to the Reviewers' comments and suggestions point by point.

Reply to the first Reviewer's report

Authors have extensively and convincingly replied to my comments. Especially, the comparison between 1T-TaS₂, 1T-TaSe₂ and 1T-NbSe₂ (this work) in light of different orbital textures is interesting and relevant. Next, I find the discussion of the possible QSL states also interesting. By performing experiments on different substrates indeed helps to elucidate the best QSL candidate state.

In conclusion, the present submission and the extensive reply fully meet the high standards of Nature Communication journal. Therefore, I recommend this submission for a publication.

Reply: We appreciate the Reviewer for the positive comment “*Authors have extensively and convincingly replied to my comments. Especially, the comparison between 1T-TaS₂, 1T-TaSe₂ and 1T-NbSe₂ (this work) in light of different orbital textures is interesting and relevant. Next, I find the discussion of the possible QSL states also interesting. By performing experiments on different substrates indeed helps to elucidate the best QSL candidate state*”. We also appreciate the Reviewer for the recommendation “*In conclusion, the present submission and the extensive reply fully meet the high standards of Nature Communication journal. Therefore, I recommend this submission for a publication*”.

Reply to the second Reviewer's report

The author emphasizes the differences from the bulk, but it seems to me that this is a repetition of what has already been reported. On the other hand, given that certain discussions have been added, it would be worthwhile to publish it and solicit a wide range of opinions.

Reply: We thank the Reviewer for the careful consideration of our manuscript and the recommendation “*given that certain discussions have been added, it would be worthwhile to publish it and solicit a wide range of opinions*”.

Reply to the third Reviewer's report

The authors have addressed some of the issues that I raised in the previous report. However, the authors' replies on the following three issues are not convincing enough.

Reply: We thank the Reviewer for the positive comment “*The authors have addressed some of the issues that I raised in the previous report*”. We really regret that our previous replies have not sufficiently addressed all the comments. We have carefully revised our manuscript according to the Reviewer’s valuable suggestions and comments, which are of great help for improving the quality of our work. We hope our response below can satisfactorily address all the comments.

(1) The results on Kondo-like resonance and the incommensurate electronic state modulations are still overemphasized. Even if there are potential differences

between 1T-NbSe₂ and 1T-TaSe₂, the presented results do not give any insight into the differences. Again, I recommend that the authors shorten (or even omit) the discussion on the Kondo resonance and the incommensurate modulation and perform more systematic experiments and theoretical analyses on the effects of magnetic molecules.

Reply: We thank the Reviewer for the valuable suggestion. We agree with the Reviewer that the results on Kondo-like resonance and the incommensurate electronic state modulations in monolayer 1T-NbSe₂ are similar to that reported in monolayer 1T-TaSe₂. According to the Reviewer's suggestion, the discussion on the Kondo resonance has been shortened to one paragraph on Page 6, and the discussion on the incommensurate modulation has been shortened to two paragraphs on Pages 6 and 7.

Moreover, we carry out the STS maps of an individual non-magnetic ZnPc molecule on monolayer 1T-NbSe₂ acquired at the band edge energies. As shown in Supplementary Figs. 14-16, there is no additional resonance peak appearing around the non-magnetic ZnPc molecule near the UHB edge, regardless of the ZnPc molecule locating on or off a SOD motif of monolayer 1T-NbSe₂ (ZnPc-top or ZnPc-hollow). This is due to the absence of spin exchange between the non-magnetic ZnPc molecule and the itinerant spinons, since the ZnPc molecule hosts no local spin. Therefore, our results highlight that the magnetic impurity is a core cause for generating the spinon Kondo scenario.

According to the Reviewer's suggestion, one paragraph has been added on Page 8 and another paragraph has been added on Page 10, in order to demonstrate the effects of a magnetic or non-magnetic molecule on the electronic properties of monolayer 1T-NbSe₂. Supplementary Figs. 15 and 16 have also been added to show the spatially resolved charge intensities of an individual non-magnetic ZnPc molecule locating on and off SOD motifs of monolayer 1T-NbSe₂ acquired at the energy of 0.65 eV, respectively.

Supplementary Fig. 15 | Spatially resolved charge intensities near the UHB edge for the ZnPc-top configuration. a,b, Topographic STM image of an individual non-magnetic ZnPc molecule adsorbed on a SOD motif. c,d, Corresponding STS map at the resonance peak energy of 0.65 eV, reflecting the absence of the spinon Kondo effect.

Supplementary Fig. 16 | Spatially resolved charge intensities near the UHB edge for the ZnPc-hollow configuration. a,b, Topographic STM image of an individual non-magnetic ZnPc molecule adsorbed off a SOD motif. c,d, Corresponding STS map

at the resonance peak energy of 0.65 eV, reflecting the absence of the spinon Kondo effect.

- (2) The authors suggested that the MnPc molecule could change the potential profile. Although I understand that it is difficult to quantitatively explain the effect of a modified potential profile, the authors should qualitatively explain why only one UHB appears instead of UHB₁ and UHB₂ and why the apparent Mott gap increases on the MnPc molecule.

Reply: We thank the Reviewer for the valuable suggestion and also appreciate the Reviewer's understanding that it is difficult to quantitatively solve the MnPc molecule on 1T-NbSe₂. According to the Reviewer's suggestion, in the following, we will first qualitatively analyze the evolution of the UHB states induced by a MnPc molecule, and then, experimentally demonstrate that the UHB states are quite sensitive to the local environment in monolayer 1T-NbSe₂.

The UHB₁ and UHB₂ states in pristine monolayer 1T-NbSe₂ are originated from the spatially varying Coulomb potential with the local maximum and minimum locating at the center and hollow sites of SOD motifs, respectively, as demonstrated in Ref. [*Nat. Phys.* 16, 218 (2020)] and our last reply (Fig. R2). A MnPc molecule has the size larger than a SOD motif, which exhibits a spatial spread over an entire SOD as it is deposited onto monolayer 1T-NbSe₂ (see Fig. 3d,e). It is possible that the Coulomb potential modification introduced by a MnPc molecule smears the original Coulomb potential difference at the center and the hollow sites of SOD. In such a case, a MnPc molecule is expected to suppress the spatially varying Coulomb potential in pristine 1T-NbSe₂ within the SOD unit cell, and the original UHB₁ and UHB₂ states in the pristine 1T-NbSe₂ may recombine into one UHB state.

In a Mott insulator, the size of the Mott gap is usually thought to be determined by the onsite Coulomb repulsion. Since a MnPc molecule induces the redistribution of the Coulomb potential within the SOD unit cell, it is possible that the resulting

effective Coulomb repulsion in the SOD unit cell gets enhanced, and hence, the size of the Mott gap measured from a MnPc molecule on 1T-NbSe₂ seemingly increases and the UHB state appears in a higher energy than that of the original UHB₁ and UHB₂ states.

Experimentally, we find out that the UHB states are quite sensitive to the local environment in 1T-NbSe₂. Here we introduce atomic defects in monolayer 1T-NbSe₂ on purpose, which can be visualized by heterogeneous SOD motifs in the STM contrast (Supplementary Fig. 20a). From the spatially resolved STS spectra shown in Supplementary Fig. 20b,c, we can find out that the energies and intensities of LHB & VB, UHB₁, and UHB₂ exhibit strongly site-dependent features. Moreover, the UHB₁ and UHB₂ can shift in energy or even vanish at specific locations (also see Ref. [*Sci. Adv.* 7, eabi6339 (2021)] for the defect-induced UHB vanishing). Therefore, our results point out that the UHB₁ and UHB₂ in monolayer 1T-NbSe₂ are sensitive to the local environment, and hence, are reasonable to be tuned by local impurities (MnPc molecule in our case).

In the revised manuscript, several sentences have been added on Pages 9 and 10 to state that the one UHB state and seemingly larger Mott gap is due to the MnPc-modified Coulomb potential profile. Supplementary Fig. 20 has also been added to demonstrate that the UHB states are quite sensitive to the local environment in 1T-NbSe₂.

Supplementary Fig. 20 | Spatially resolved STS spectra of heterogeneous monolayer 1T-NbSe₂. a, Typical STM image of monolayer 1T-NbSe₂ with atomic defects, exhibiting enhanced or weakened SOD spots. b, Spatially resolved STS spectra recorded along the red arrow marked in panel a. The energies and intensities of LHB & VB, UHB₁, and UHB₂ are strongly changed, as marked by the blue, red, and yellow dashed lines, respectively. c, Typical STS spectra extracted from panel b. The UHB₁ and UHB₂ can shift in energy or even vanish at different locations.

- (3) The authors claim that non-magnetic ZnPc molecules do not produce spinon Kondo peaks. In Supplementary Fig. 14 f, I see a shoulder just below the UHB that is similar to the putative spinon Kondo peak on the MnPc.

Reply: We thank the Reviewer for the valuable comment. The shoulder-like feature generated by a non-magnetic ZnPc molecule shown in Supplementary Fig. 14f is much less obvious than that of magnetic MnPc molecule shown in Fig. 4. More importantly, the STS maps of the ZnPc-hollow configuration (Supplementary Fig. 14f is related to the ZnPc-hollow configuration) acquired near the UHB edge of 0.65 eV exhibit no charge density concentrating near the ZnPc (Supplementary Fig. 16). This phenomenon is quite different from that of the magnetic MnPc molecule

where the electrons at the energy near the UHB edge of 0.65 eV mainly locate around the MnPc in the MnPc-hollow configuration (Supplementary Fig. 13). Therefore, our results unambiguously demonstrate that the non-magnetic ZnPc molecule cannot generate the spinon Kondo peak.

In our revised manuscript, Supplementary Fig. 16 has been added to show the spatially resolved charge intensities of an individual non-magnetic ZnPc molecule locating off SOD motifs of monolayer 1T-NbSe₂ acquired at the energy of 0.65 eV. Several sentences have also been added on Pages 8 and 10 to describe the experimental phenomena, according to the Reviewer's comment.

Supplementary Fig. 13 | Spatially resolved intensities of the resonance peak appearing near the UHB edge for the MnPc-hollow configuration. a,b, Topographic STM image of a MnPc molecule adsorbed off a SOD motif. c,d, Corresponding STS map at the resonance peak energy of 0.65 eV, reflecting the spatial variations of the spinon Kondo effect.

List of changes:

1. On Page 6, the discussion on the Kondo resonance has been shortened to one paragraph.
2. On Pages 6 and 7, the discussion on the incommensurate modulation has been shortened to two paragraphs.
3. On Page 8, the sentences “For comparison, we also carry out similar STM and STS measurements of a non-magnetic molecule ZnPc on monolayer 1T-NbSe₂, as summarized in Supplementary Figs. 14-16. From the site-dependent STS spectra shown in Supplementary Fig. 14, we can find out that there is almost no obvious resonance peaks emerging at the band edges when recorded on the central Zn ion, regardless of the ZnPc molecule locating on or off a SOD motif of monolayer 1T-NbSe₂. Our spatially resolved STS maps acquired at the band edge energies further reveal the absence of additional resonance peaks around the non-magnetic ZnPc molecule (Supplementary Figs. 15 and 16), which exhibits significant difference from those of the magnetic MnPc molecule (Supplementary Figs. 12 and 13). These phenomena highlight that the magnetic impurity is a core cause for generating the additional resonance peaks in monolayer 1T-NbSe₂” have been added to describe the electronic properties of an individual non-magnetic ZnPc molecule on monolayer 1T-NbSe₂.
4. Supplementary Figs. 15 and 16 have been added to show the spatially resolved charge intensities of an individual non-magnetic ZnPc molecule locating on and off SOD motifs of monolayer 1T-NbSe₂ at the energy of 0.65 eV, respectively.
5. On Pages 9 and 10, the sentences “From our experiments we can find out that the original UHB₁ and UHB₂ undergo a spectral weight re-distribution under the MnPc-modified Coulomb potential profile, resulting in the UHB₁ suppressed and the UHB₂ merging into a higher energy. Although we cannot quantitatively explain such a phenomenon, we believe UHB states are quite sensitive to the local environment because atomic defects in monolayer 1T-NbSe₂ can also influence the energies and intensities of the UHB₁ and UHB₂ in a similar way

(Supplementary Fig. 20)” have been added to demonstrate the sensitivity of the UHB states to the local impurities.

6. Supplementary Fig. 20 has been added to show the modification of the UHB₁ and UHB₂ in monolayer 1T-NbSe₂ by atomic defects.
7. On Page 10, the sentences “However, for a non-magnetic ZnPc molecule with $S = 0$ spin, there is no spin exchange between the ZnPc molecule and the itinerant spinons, since a ZnPc molecule hosts no local spin. As a result, the additional resonance peaks are not expected to appear at the band edge of monolayer 1T-NbSe₂, which is in well consistent with our measurements shown in Supplementary Figs. 14-16. Therefore, the observed site-dependency of the resonance peaks generated by a magnetic MnPc molecule, complementary by the absence of resonance peaks generated by a non-magnetic MnPc molecule, are unambiguously demonstrate that the additional resonance peaks at the band edges are attributed to the spinon Kondo scenario of a magnetic impurity deposited on a U(1) QSL with SFS” have been added to describe the spinon Kondo scenario with magnetic and non-magnetic impurities.

REVIEWERS' COMMENTS

Reviewer #3 (Remarks to the Author):

I appreciate the authors' revisions, which address most of my concerns in the previous versions. I think that more experiments, especially systematic changes of the band-edge resonances in other metal phthalocyanine molecules with different spins ($1/2$, $5/2$...), should be done to come to the "conclusive evidence". Nevertheless, I agree that the authors' work provides a new way to investigate the quantum spin liquid. I am happy to recommend the publication of the revised manuscript in Nature Communications.

Re: Manuscript ID: NCOMMS-23-36519B

Title: Quantum spin liquid signatures in monolayer 1T-NbSe₂

Reply to the third Reviewer's report

I appreciate the authors' revisions, which address most of my concerns in the previous versions. I think that more experiments, especially systematic changes of the band-edge resonances in other metal phthalocyanine molecules with different spins (1/2, 5/2...), should be done to come to the "conclusive evidence". Nevertheless, I agree that the authors' work provides a new way to investigate the quantum spin liquid. I am happy to recommend the publication of the revised manuscript in Nature Communications.

Reply: We appreciate the Reviewer for the positive comments "*I appreciate the authors' revisions, which address most of my concerns in the previous versions*" and "*I agree that the authors' work provides a new way to investigate the quantum spin liquid*". We also appreciate the Reviewer for the recommendation "*I am happy to recommend the publication of the revised manuscript in Nature Communications*".

We agree with the Reviewer that it is better to carry out more experiments on metal phthalocyanine molecules with different spins. However, it is failed after many attempts, owing to the limitation of our experimental conditions. Instead, the relevant theoretical predictions are presented in our manuscript, and the experiments need further exploration.